# Continuous and scalable manufacture of amphibious energy yarns and textiles

Wei Gong[1], Chengyi Hou[1], Jie Zhou[2], Yinben Guo[1], Wei Zhang[1], Yaogang Li[3], Qinghong Zhang[3] & Hongzhi Wang[1]

Biomechanical energy harvesting textiles based on nanogenerators that convert mechanical energy into electricity have broad application prospects in next-generation wearable electronic devices. However, the difficult-to-weave structure, limited flexibility and stretchability, small device size and poor weatherability of conventional nanogenerator-based devices have largely hindered their real-world application. Here, we report a highly stretchable triboelectric yarn that involves unique structure design based on intrinsically elastic silicone rubber tubes and extrinsically elastic built-in stainless steel yarns. By using a modified melt-spinning method, we realize scalable-manufacture of the self-powered yarn. A hundred-meter-length triboelectric yarn is demonstrated, but not limited to this size. The triboelectric yarn shows a large working strain (200%) and promising output. Moreover, it has superior performance in liquid, therefore showing all-weather durability. We also show that the development of this energy yarn facilitates the manufacturing of large-area self-powered textiles and provide an attractive direction for the study of amphibious wearable technologies.

[1] State Key Laboratory for Modification of Chemical Fibers and Polymer Materials, College of Materials Science and Engineering, Donghua University, 201620 Shanghai, P.R. China. [2] College of Electronics and Information Engineering, Sichuan University, 610064 Chengdu, P.R. China. [3] Engineering Research Center of Advanced Glasses Manufacturing Technology, Ministry of Education, Donghua University, 201620 Shanghai, P.R. China. Correspondence and requests for materials should be addressed to C.H. (email: hcy@dhu.edu.cn) or to Q.Z. (email: zhangqh@dhu.edu.cn) or to H.W. (email: wanghz@dhu.edu.cn)

With the rapid development of wearable electronic devices, there is a greater demand for wearable energy harvesting. Due to the lack of the flexibility, comfort and lightness, conventional bulky batteries may be replaced by yarn- or fabric-based flexible energy devices[1–4]. Lithium-ion batteries[5,6] and electrochemical supercapacitors[7,8] have been integrated into textiles, but these flexible energy devices require frequent and inconvenient charging, which largely hinders the practical, sustainable, and broad-range applications of the wearable electronics. Electromagnetic generators suffer from high cost per watt when scaled to the millimeter and smaller dimensions needed for emerging applications[9]. Piezoelectric and ferroelectric generators lack the elasticity needed for harvesting large strains[10]. Solar cells are available only in the daytime[11]. Hence, triboelectric nanogenerators (TENGs) that convert mechanical energy into electricity have received extensive attention owing to capability as a continuous, self-sufficient and sustained power-supplying source[12,13]. In particular, it shows clear advantages for harvesting low-frequency and irregular mechanical energy such as human motion[14].

The application of the burgeoning TENG technology in the traditional textile industry offers more potential for energy textiles (e-textiles). However, the development of biomechanical energy harvesting textiles still faces some big challenges. First, scalable manufacture of TENGs is still very immature. TENGs have fine and complicated structures that makes continuous fabrication difficult. The existing centimeter-length and hand-made devices offer limited possibility to promote in the e-textile industry[15–22]. Second, the performance of the TENG may be greatly suppressed by high humidity and liquid contact[23–25]; therefore, wet environments may be detrimental to TENG-based textile. Although several works have reported underwater-operational TENGs[26–31], they sacrifice flexibility and wearability. In addition, the existing energy harvesting textiles are mostly fabricated by weaving (knitting, or braiding) at least two kinds of fibers (or yarns) that hold opposite electrical properties[32–35]. The individual raw fibers do not show TENG behaviors. Besides, they are susceptible to humidity, and require more stringent weaving techniques in order to achieve better wearability and stretchability.

Herein, we report a continuous and scalable method for spinning an underwater-operational single-electrode triboelectric yarn (SETEY). The SETEY consists of intrinsically elastic silicone rubber tubes and built-in helical-structure stainless steel yarns, showing super stretchability and flexibility. Besides, it shows high performance in both air and liquid. The SETEY-weaved energy textiles are stretchable, washable, and all-weather available for harvesting biomechanical energy and monitoring motion signals.

## Results

### Continuous and scalable manufacture of the single-electrode triboelectric yarns

The core-sheath yarn structure is fabricated through blow-molding silicone rubber tube over the surface of stainless steel yarn using a specialized spinning equipment (Fig. 1a, b, Supplementary Note 1, and Supplementary Figs. 1 and 2). The sheath and core materials are continuously stretched forward by compaction rollers I and II, respectively. The tensile force ($F_{x1}$) on the sheath material changes as it moves through pully blocks, where a controllable pre-strain ($\varepsilon$) is introduced in the sheath material, which can be calculated by:

$$\varepsilon = \frac{1}{m}\left(\frac{F_y}{n} - b\right), \tag{1}$$

where $n$ is the number of pulley blocks, $F_y$ is the resultant gravity, $m$ and $b$ are the slope and intercept obtained from the tensile force-

strain curve of the material, respectively. Details can be seen in Supplementary Fig. 3, Supplementary Note 2, and Supplementary Movie 1. The prestretched sheath material is then released due to the differential rotational speeds of compaction roller and reeling roller (Supplementary Note 3). During releasing, the friction between the sheath and core materials drives the straight core yarn into helix. The as-obtained SETEY is collected on reeling roller (Fig. 1c and Supplementary Movie 2). The formation of the built-in helix is a self-organized process cooperating the force field (pulley blocks) and the velocity field (compaction roller and reeling roller). A typical $V_3$ of 45 rpm and $r_3$ of 10 cm lead to a ~28 m min$^{-1}$ collection speed. The fabrication can be sped up by simply lengthening the production line. The built-in helix structure of core yarn as well as the elastic nature of rubber sheath (Supplementary Fig. 4 and Supplementary Note 4) enable various kinds of deformation of the SETEY, including stretching, pressing, bending, and twisting (Supplementary Fig. 5). During stretching, for example, the core and sheath deform synchronously (Supplementary Note 5, Supplementary Figs. 6 and 7), which promises stable electrical and mechanical behaviors of the unique structure. Particularly, it exhibits a large elastic strain of 200% and good durability (Supplementary Fig. 4c−e, Supplementary Fig. 8 and Supplementary Note 6), being able to satisfy the requirement for wearable applications.

### Working principle of the single-electrode triboelectric yarns

The operating pattern of the SETEY is clearly displayed in Supplementary Movie 3. The working principle of the SETEY is presented in Fig. 2a, Supplementary Fig. 9, and Supplementary Note 7. During deformation, e.g., stretching, the contact area between the sheath tube and the core yarn decreases while the surface area of the sheath tube increases, both contributing to the in-plane charge separation and alteration of the charge distribution. The increased amount of holes that distributed on the surface of the highly conductive core yarn drive electrons flow from the ground electrode into the SETEY. When the SETEY is released, electrons flow back. Alternating current can be produced via repeatedly stretching and releasing the SETEY. Though this mechanism has been roughly explained in our previous work[15], the in-depth contact mechanism of this metal-noncrystalline polymer (M-NP) system was yet understood.

For understanding the coupling effect of contact electrification and electrostatic induction, the in-depth contact mechanisms have been studied in metal−semiconductor[36], polymer−polymer[36] and metal−crystalline polymer[37] systems, which were based on the double-electrode mode. Here, according to the potential well model[36], the reasonable assumptions about the charge transfer mechanism of the single-electrode M-NP system is proposed for the first time. In the M-NP single-electrode potential well model, an atom can be represented by a potential well in which the out shell electrons are loosely bounded, forming an electron cloud of the atom or molecule. As shown in Fig. 2b (part I), the barrier between metal and noncrystalline polymer enhances the local trapping effect of the potential wells. Once the metal helix is built in the noncrystalline polymer tube, two materials contact and interface forms, where the electron clouds overlap. Consequently, two individual potential wells become one double-well to reach chemical potential equilibrium (Fig. 2b, part II). Under deformation, e.g., stretching, the separation of materials leads to the break of full contact state. At this point, in order to fill the new potential well of the metal, electrons are driven into the metal from the ground electrode (Fig. 2b, part III). At fully separated state (Fig. 2b, part IV), the barrier is regenerated and the potential wells of both materials is filled, so the system reaches a steady state. Most of the electrons transferred to noncrystalline polymer tube will be maintained

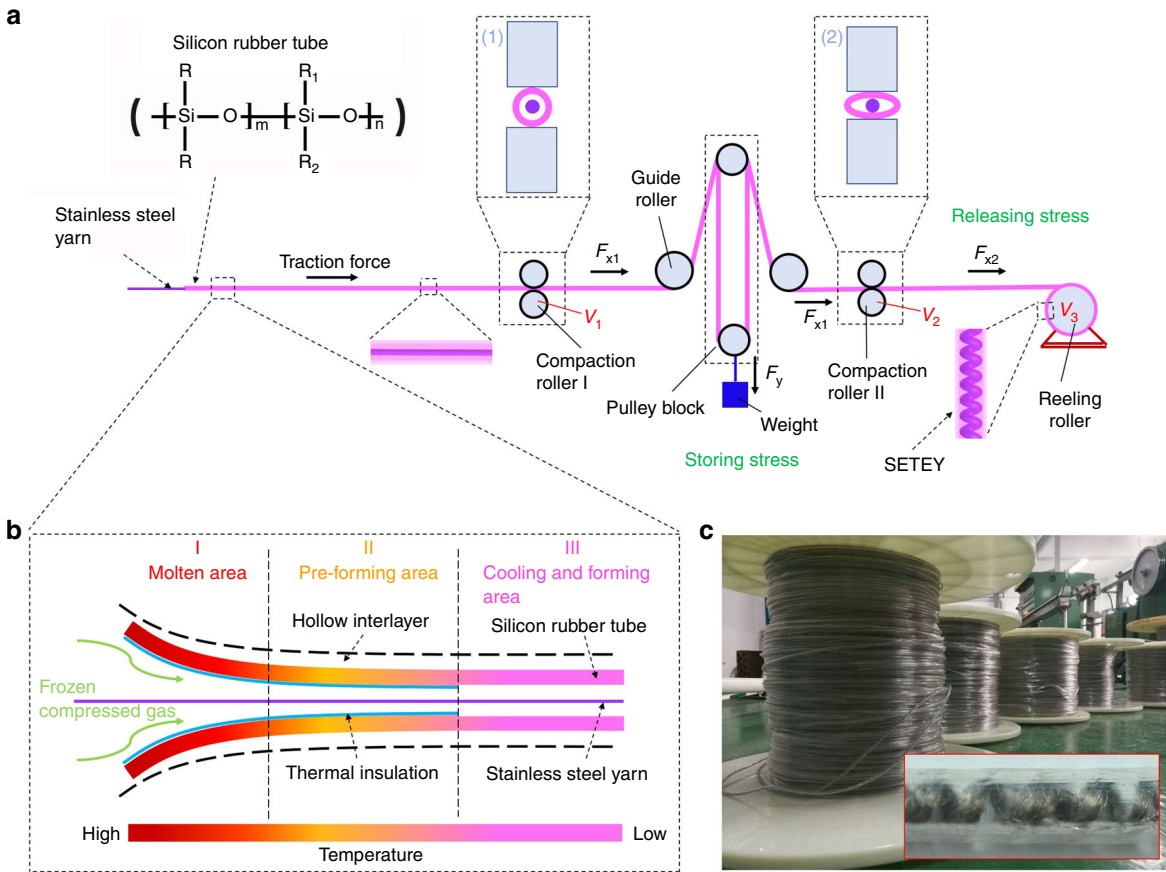

**Fig. 1** Continuous manufacture of amphibious energy yarns. **a** Schematic diagram of a specialized melt-spinning process for scalable manufacture of single-electrode triboelectric yarn (SETEY). Light pink indicates silicone rubber tube while purple indicates stainless steel yarn. $F_{x1}$ and $F_{x2}$ are the tensile force on the silicone rubber tube during the stress storing and releasing, respectively. $F_y$ is the resultant gravity. The rotational speed ($V_1$) of compaction roller I equals to that ($V_2$) of compaction roller II. The rotational speed ($V_3$) of reeling roller = $r_2V_2/(1 + \varepsilon)r_3$, where $r_2$ and $r_3$ are radius of the compaction roller II and reeling roller, respectively. $\varepsilon$ is prestrain. Insets (1) and (2) are enlarged cross-sectional views of compaction rollers I and II, respectively. **b** The enlarged view of silicone rubber tube molding process. Temperature: 250–280 °C in area I, 120–180 °C in area II, and 30–50 °C in area III. **c** The photograph of SETEY collected on reeling rollers. Inset: the enlarged view of an individual SETEY

due to the energy barrier $E_2$ present in potential well of noncrystalline polymer tube[38]. In releasing, metal and noncrystalline polymer tube contact again and produce one double well. The chemical potential needs to be restored to equilibrium and the potential well of the noncrystalline polymer tube is full, which can drive electrons from the metal return to the ground electrode. The M-NP single-electrode potential well model is also verified through numerical simulation using COMSOL, as illustrated in Fig. 2c and Supplementary Note 8. In full contact state (Fig. 2c, left panel), the positive potential on the metal and the negative potential on the noncrystalline polymer tube are fully screened owing to potential equilibrium. When the noncrystalline polymer tube is stretched outward, the electric potential on the noncrystalline polymer tube reaches its maximum because of potential separation. (Fig. 2c, right panel).

According to the above theory, the overall output performance of the SETEY is determined by the total area of double-well region and the degree of well separation during deformation. Thus, we are able to alter the SETEY performance through changing key parameters (Supplementary Fig. 10 and Supplementary Note 9), including the inner diameter of the silicone rubber tube (Supplementary Figs. 10b and 11), the diameter of the stainless steel yarn (Supplementary Fig. 10e), the length of the SETEY (Supplementary Figs. 10i and 12), the tensile strain (Supplementary Figs. 10j and 13) and frequency (Supplementary Figs. 10k, 14 and 15). All tests on the SETEY were performed for

approximately 2000 cycles in open air to ensure the long-term stability/reliability of the device (Supplementary Fig. 16). This working mechanism of the built-in helix SETEY causes it to exhibit completely different features than the common flat structure. We compared the difference between the built-in helix structure and the flat structure and found that the output performance of the former is larger and more stable (Supplementary Note 10, Supplementary Fig. 17a, and 19). Interestingly, the output voltage of the SETEY is inert to humidity change (from 20 to 90% relative humidity) and soaking in sweat (Supplementary Figs. 20 and 21). This is attributed to the waterproof core-sheath structure and the hydrophobicity of the sheath material (Supplementary Fig. 22). It makes the SETEY more adaptable to complex real-world environments in comparison to many of the state-of-the-art TENG devices[23–25].

**Amphibious energy yarns working in liquid.** Exposure to molecules that are electrical conductive or chemical reactive is a major concern for sensitive electronics. Therefore, liquid is usually bad for electronic devices. However, we find that the polarization of liquid molecules is anomalously beneficial to the SETEY. As demonstrated in Fig. 3a, the SETEY was operated in nitrogen and various liquid environments respectively. Though the core material is resistant to external molecules, the silicone rubber sheath fully contacts the gas/liquid medium. The output voltage of SETEY

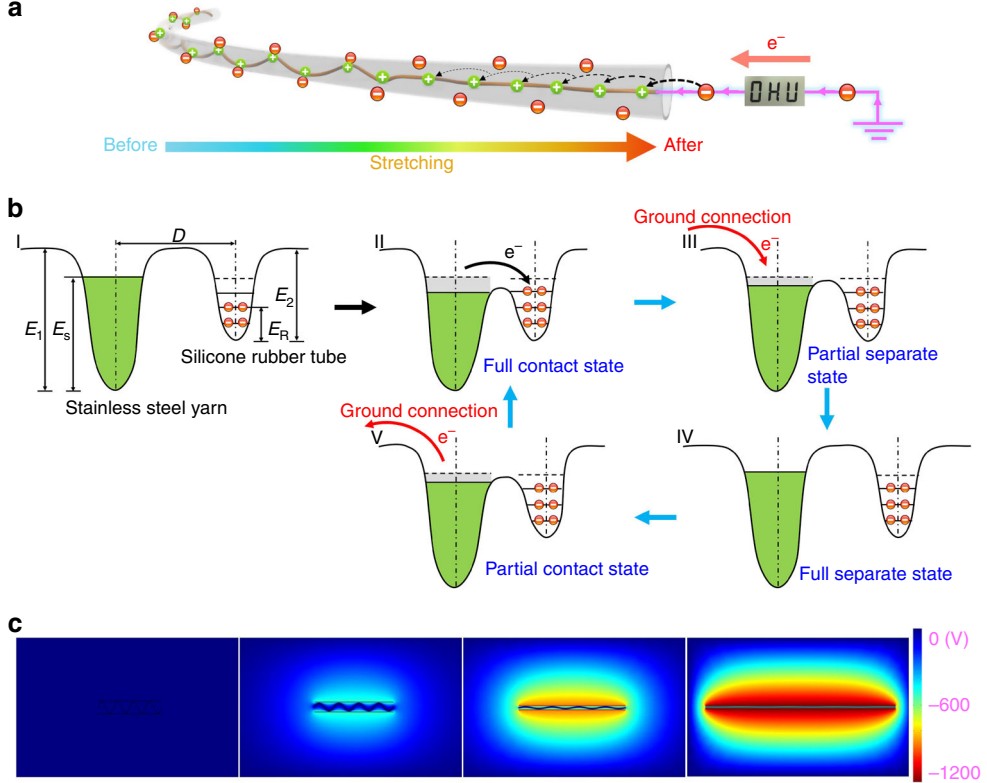

**Fig. 2** Working mechanism of the single-electrode triboelectric yarn. **a** Kinetic demonstration of the stretching process of the SETEY. **b** A potential well model proposed for explaining charge transfer during the contact and separation of two materials under a single electrode. $D$ is the distance between potential wells of inner stainless steel yarn and silicone rubber tube sheath, $E_S$ and $E_R$ are the occupied energy levels of electrons in the atoms of stainless steel yarn and silicone rubber tube, $E_1$ and $E_2$ are the required potential energies for electrons to escape from the surfaces of stainless steel yarn and silicone rubber tube, respectively. $E_S$ and $E_R$ are respectively smaller than $E_1$ and $E_2$. **c** Simulation results of the potential distribution of the SETEY by using COMSOL software. From left to right: the potential on the silicone tube gradually increases and the potential on the stainless steel yarn remains stable. SETEY single-electrode triboelectric yarn

increases with the increase of dielectric constant of environmental molecules. Media with relative low and high dielectric constant ($N_2$ and water, respectively) are further applied to SETEYs with different sheath thickness. Interestingly, the output voltage measured in $N_2$ remains unchanged but it increases remarkably in water with decreasing SETEY sheath thickness (Fig. 3b). The above results strongly indicate a potential-polarization coupling effect occurred on the surface of the SETEY. Detailed explanations are shown in Supplementary Fig. 24 and Supplementary Note 11. In Fig. 3c, we illustrate the dynamic process of polarization of water molecules occurred along with potential generation of the SETEY. The surface potential from triboelectrification will induce polarization of surrounding bulk water, while the latter will enhance the former in subsequent contact electrification processes until an equilibrium is reached. This coupling effect will greatly enhance the amount of triboelectric charges that can be generated by SETEY. A detailed analysis of the dynamic polarization process is shown in Supplementary Fig. 25 and Supplementary Note 12. Accordingly, large molecular polarity and strong electrostatic field are both positively correlated to the triboelectric effect, as already demonstrated in Fig. 3a, b, Supplementary Fig. 24, and Supplementary Note 11. Besides, Fig. 3a also shows an interesting relationship between the device output and relative dielectric constant of environmental medium. The growth trend of the output voltage of the SETEY gradually decreases as the relative dielectric constant increases (Supplementary Note 13 and Supplementary Fig. 26). Overall, the SETEY shows superior performance in water than in air, which is a newly discovered and counterintuitive phenomenon. To some extent, the potential-polarization coupling effect is similar to the

surface-dielectric polarization effect between triboelectrification surface and ferroelectric material[39].

It is shown that the SETEY can power a liquid crystal display (LCD) in water (Fig. 3d and Supplementary Movie 4). Its function is reliable in shallow water (Fig. 3e). Thus, the SETEY is useful as an amphibious wearable power supply.

**Electrical output performance of the energy textiles**. To further explore the real-world applications of the SETEY, we integrate it into textiles. Though biomechanical energy harvesting textiles have shown promising potential in wearable energy systems[35,40,41], challenges in industrial production of e-textile still need to be addressed. Besides, short circuit that may be caused under various wet conditions is another major concern for e-textiles. The SETEYs can be woven with itself or with other yarns. Since the latter performs better owing to higher output and better practicality (Supplementary Fig. 27 and Supplementary Note 14), we will discuss the latter results next. Herein, we show that owing to the ultra-flexibility (Supplementary Fig. 28 and Supplementary Note 15) and unlimited length of the SETEY, we are able to use a braiding machine to continuously weave e-textiles using SETEY and other commercial yarns. Moreover, the employ of SETEY solves the short-circuit problem. Figure 4a illustrates the fabrication procedures of the SETEY-based e-textiles. Commercial stainless steel yarn and water-resistant modified polyacrylonitrile yarn (Supplementary Fig. 29) is twisted to form a mechanically stable double-plied yarn (Supplementary Fig. 30). Couple moment ($M_0$) ensures the flexibility of the

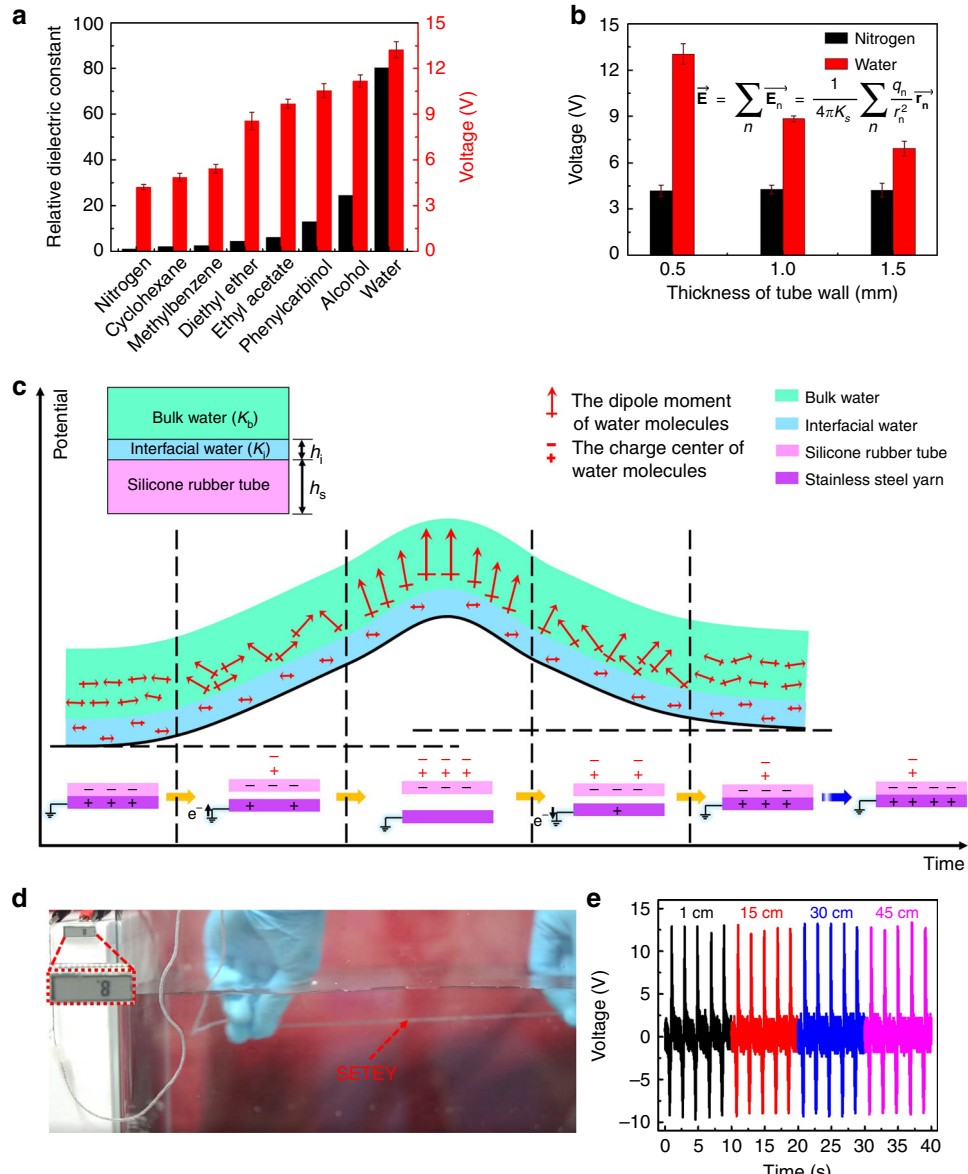

**Fig. 3** Working mechanism of amphibious energy yarns during underwater operation. **a** The output voltages of the single-electrode triboelectric yarn (SETEY) in nitrogen, cyclohexane, methylbenzene, diethyl ether, ethyl acetate, phenylcarbinol, alcohol and water. A SETEY with sheath thickness of 0.5 mm and length of 30 cm was tested in these experiments. The error bars correspond to standard deviation caused by the statistical uncertainty of measurement. **b** The output voltages of SETEYs with different sheath thickness measured in nitrogen and water. The principle of field intensity superposition $\left( \vec{\mathbf{E}} = \sum_n \overrightarrow{\mathbf{E_n}} = \frac{1}{4\pi K_s} \sum_n \frac{q_n}{r_n^2} \overrightarrow{\mathbf{r_n}} \right)$ is used to describe the relationship between the electric field intensity at different locations and its distance to the point charge group. $\overrightarrow{\mathbf{E_n}}$ is the electric field intensity of a point charge at a certain point, which is a vector. $q_n$ is the charge of a point charge. $\overrightarrow{\mathbf{r_n}}$ is the radial vector from a certain point in the electric field to a point charge. $K_s$ is the relative dielectric constant of the silicone rubber. $\vec{\mathbf{E}}$ is the total electric field intensity at a certain point under the point charge group. The error bars correspond to standard deviation caused by the statistical uncertainty of measurement. **c** Schematic of a dynamic polarization process during single cyclic tensile of the SETEY in water. Water exhibits distinct layered structures near surface of the silicone rubber tube. $h_i$ and $h_s$ are the thicknesses of the interfacial water and the silicone rubber wall, respectively. $K_i$ and $K_b$ are dielectric constants of the interfacial water and the bulk water, respectively. $K_i$ is only ~2 [42,43] while $h_i$ is only 1.5−2 nm; therefore, interfacial water is difficult to reorient in electric field[44−47]. The surface potential from triboelectrification only interacts with bulk water ($K_b \approx 80$). **d** Digital photos showing the self-powered application: A liquid crystal display (LCD) is lit up by the SETEY in water. **e** The output performances of the SETEY measured under different depth in water

multi-level structure while the resultant torsional moment strengthens the structure to avoid nonuniform tension (see Fig. 4b, Supplementary Fig. 31 and Supplementary Note 16). Meanwhile, the stainless steel yarn offers extremely high conductivity (830 S cm$^{-1}$); thus, it can act as an outer electrode. This double-plied yarn is weaved with SETEY. The weaving process

and key parameters are also shown in Supplementary Fig. 32, Supplementary Note 17, and Methods.

Schematic illustration and digital photograph of the e-textile are illustrated in Supplementary Fig. 33. Corresponding to different mechanical deformations, the operation mode of the e-textile can be divided into stretch and compress modes (Supplementary Fig. 34

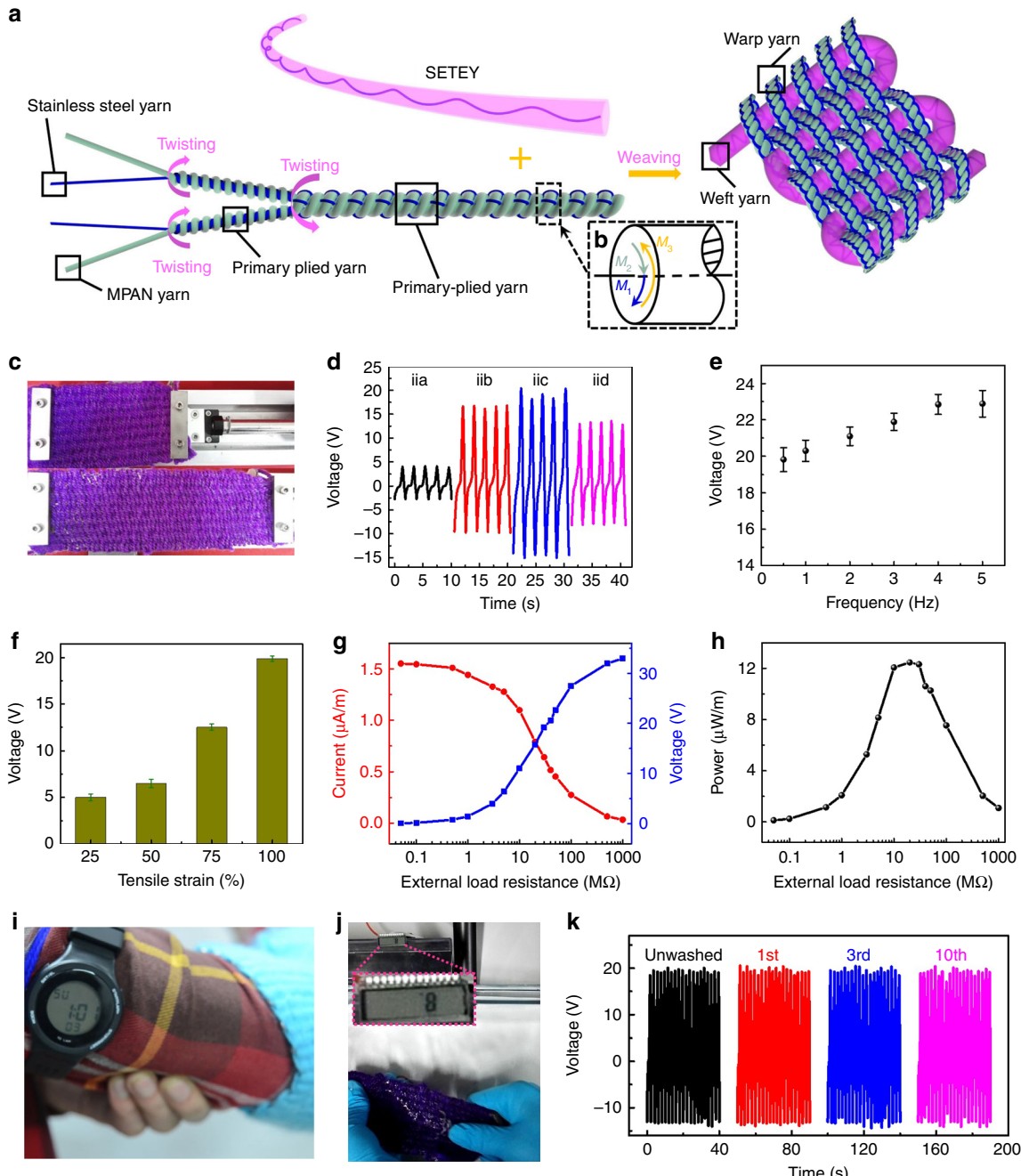

**Fig. 4** Structure design and electrical output performance of the energy textiles. **a** Fabrication processes of the e-textile. **b** Resultant torsional moment diagram of the double-plied yarn. $M_1$ and $M_2$ are the torsional moments of the two primary plied yarn, respectively. $M_3$ is the torsional moment of the double-plied yarn. Their resultant torsional moment is 0, namely $M_3 = M_1 + M_2$. **c** Photographs of the experiment setup with a stretch-retraction mode. **d** Electrical outputs of four distinct patterns measured under 100% tensile strain at a fixed frequency of 0.5 Hz. **e** Output voltage of the warp-weft-connection single-electrode e-textile measured at different frequency (0.5–5 Hz) with a 100% tensile strain. The error bars correspond to standard deviation caused by the measurement noise. **f** Output performance of the warp-weft-connection single-electrode e-textile measured under different tensile strain (25, 50, 75 and 100%). The error bars correspond to standard deviation caused by the statistical uncertainty of measurement. **g** Output current/voltage, and **h** power of the warp-weft-connection single-electrode e-textile measured at different external load resistances varied from 50 KΩ to 1 GΩ with a fixed frequency of 0.5 Hz. **i** An image of a self-charging system that harvests biomechanical energy by the e-textile to power an electronic watch. **j** A snapshot of an e-textile working underwater. **k** The electrical output results of the e-textile after repeated washing

part ii and 34 part iii, respectively). Each mode has four types of circuit connection patterns (Supplementary Figs. 34a, b, and 35), which directly affects their mechanical-to-electrical conversion performances. The e-textile can also be categorized based on circuit connection patterns, i.e. warp-connection single-electrode pattern (Supplementary Figs. 34a and 35a), weft-connection single-electrode pattern (Supplementary Figs. 34b and 35b), warp-weft-connection

single-electrode pattern (Supplementary Figs. 34c and 35c), and double-electrode pattern (Supplementary Figs. 34d and 35d). In the following, we discuss the electrical output performances of the e-textile from two aspects, i.e., the operation mode and the circuit connection pattern.

The working mechanism of the warp-weft-connection single-electrode e-textile under normal circumstances is briefly

demonstrated in Supplementary Fig. 36a, which is based on a conjunction of triboelectrification and electrostatic induction. A detailed analysis is shown in Supplementary Figure 36 and Supplementary Note 18. The potential distribution of other three circuit connection patterns is shown in Supplementary Fig. 37. When the e-textile is short-circuited in water, the double-plied yarn that is in direct contact with water is at zero potential, leading to electrode pattern shift in e-textile: from warp-weft-connection single-electrode pattern to weft-connection single-electrode pattern (Supplementary Fig. 38a). The auto-shift operation modes enable the e-textile to work amphibiously. As shown in Supplementary Fig. 38b, the output performance of the e-textile of the weft-connection single-electrode pattern under-water is superior to that in the air owing to the potential-polarization coupling effect, and output performance of the e-textile is also stable during the tensile test.

The special weaving structure of the e-textile also enables high stretchability. As illustrated in Fig. 4c, a linear motor is used to stretch an e-textile woven by a roughly 1-m-long SETEY weft yarn and a double-plied wrap yarn. The electrical outputs of the four distinct patterns are shown and compared under 100% tensile strain at a given frequency of 0.5 Hz and an external load of 20 MΩ (Fig. 4d). The warp-weft-connection single-electrode e-textile shows best output performance among different circuit connection modes. The performance analysis of other three circuit connection patterns can be seen in Supplementary Note 19. We focus rationally on the warp-weft-connection single-electrode in the following. As shown in Fig. 4e, when the stretching frequency varies from 0.5 to 5 Hz, the output voltage of the warp-weft-connection single-electrode e-textile increases from 19.8 to 22.9 V. In other words, the increase of frequency is favorable for the magnitude of output voltage. Figure 4f shows the output voltage of the e-textile measured under applied tensile strains of 25, 50, 75, and 100% at a frequency of 0.5 Hz with an external load of 20 MΩ. The output voltage increases with increasing the tensile strain. At a fixed contact frequency of 0.5 Hz, with the increasing of the applied external resistances, the maximum output current drops, while the maximum output voltage follows a reverse trend, which is due to the Ohm's law (Fig. 4g). The instantaneous power is maximized at a load resistance of 20 MΩ, corresponding to a peak power of about 12.5 μW m$^{-1}$ (Fig. 4h). The power is calculated as $I^2R$, where $I$ is the output current across the external load and $R$ is the load resistance. In addition, compression performances of the four circuit connection patterns are shown in Supplementary Fig. 39 and Supplementary Note 20.

As a wearable device, the energy textile is able to harvest mechanical energy from tensile and compressive body motions. As demonstrated in Fig. 4, part i and Supplementary Movie 5, the mechanical energy harvesting textile could sustainably drive an electronic watch. The as-generated electrical energy can also be regulated to charge capacitors and to power different portable electronics, as illustrated in Supplementary Fig. 40, Fig. 4j and Supplementary Movie 6 demonstrate that the energy textile is underwater operational in its warp-weft-connection single-electrode pattern. Owing to that the double-plied structure of the warp yarn can effectively resist complex mechanical agitation and internal deformation, the shrinkage of both yarn and textile is barely observed after long-term operation under water and even after laundering. In a rigorous laundering testing, the energy textile was laundered for 30 min using the standard procedure of a commercial washing machine (see Methods). After drying naturally, the electrical output of the energy textile was measured. There is no decay in its performance after ten laundering tests (Fig. 4k).

Finally, as a real example, we show that our energy textiles are integrated into a wireless monitoring system for real-time monitoring of body movements (Supplementary Fig. 41, Supplementary Movie 7 and Supplementary Note 21). The energy harvesting textiles can generate varying electrical signals with the elbow bending, which are then wirelessly transmitted to the cell phone via Bluetooth. Meanwhile, the energy textiles show potential to charge Li-ion batteries of the wireless monitoring system. As a wearable self-powered active kinematic sensor or power supply, the scalable-manufactured energy yarns and textiles are readily working, while there is still space for further improvement as for the devices electrical output.

## Discussion

In summary, we develop and apply a spinning technology for scalable manufacture of triboelectric yarns, which integrates only commercial and low-cost materials: silicone rubber and stainless steel yarn. We discussed the in-depth mechanism of its mechanical-to-electrical energy conversion behavior and revealed its capability of underwater operation and large deformation. Weaving this energy yarn with other commercial yarns, we develop a mechanically stable, highly stretchable, and large-scale energy textile.

## Methods

**Preparation of the energy textile**. The e-textile was woven with SGA598 Braiding Machine (Jiangyin Tongyuan-fj Co., Ltd.). First of all, we adjusted the position of the heald frame. Then setting the parameters on the writing board screen: the working air pressure of the braiding machine was ~0.4–0.6 MPa and picks per minute were ~50–70 times. Afterward, the yarn after warping was fixed at the rear of the heald frame with a yarn guide for drawing-in. Finally, the weft yarn was machined and the tension of the warp and weft yarns was adjusted by the back beam during the weaving process.

**Characterization and measurements**. Mechanical properties of the silicone rubber tube and the SETEY were measured with an electronic universal testing machine (model no. Instron 5969, from Instron Corporation). For the measurement of the electrical output capability of the TENG, external forces were applied by a mechanical motor (homemade equipment), which corresponded to the stretching and compressing operations, respectively. The contact angles of droplet were tested using a contact angle goniometer (Kino SL200B, USA) equipped with a tilting base. The RH was monitored with a hygrometer. The electrical outputs were recorded by an oscilloscope (LeCroy, Wavesurfer 104MXs-B, USA) and an electrochemical workstation (CHI760D, Shanghai Chenhua Instruments, China). Optical microscope photograph of the double-plied yarn was measured by Leica DVM6.

**Washing test**. The e-textile was put into a commercial laundering machine (EG10014B39GU1, Haier). Each laundering cycle lasted for 30 min, and the agitator started to rotate at a spinning speed of 120 rpm for 10 min. After running each cycle and drying naturally, the electrical outputs after washing at different times were measured.

## Data availability

The data that support the findings of this study are available from the corresponding authors upon reasonable request.

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

## Acknowledgements

We gratefully acknowledge the financial support by the Fundamental Research Funds for the Central Universities (2232019A3-02), DHU Distinguished Young Professor Program (LZB2019002), Natural Science Foundation of China (61775131), Science and Technology Commission of Shanghai Municipality (16JC1400700), Innovation Program of Shanghai Municipal Education Commission (2017-01-07-00-03-E00055), Program of Shanghai Academic Research Leader (16XD1400100), and the Program of Introducing Talents of Discipline to Universities (No. 111-2-04). C.H. thanks the Shanghai Natural Science Foundation (16ZR1401500), the Shanghai Sailing Program (16YF1400400), and Young Elite Scientists Sponsorship Program by CAST (2017QNRC001). W.G. thanks the Donghua University Doctoral Innovation Fund Program.

## Author contributions

W.G. and C.H. conceived the idea, designed the experiment and guided the project. W.G., Y.G. and W.Z. performed the experiments and measurements. J.Z. performed COMSOL simulation. Y.L. revised the manuscript. W.G., C.H., Q.Z. and H.W. analyzed the experimental data, drew the figures and prepared the manuscript. All authors discussed the results and reviewed the manuscript.
