## [Peer Review File · Nature Communications]

Reviewers' comments:

Reviewer #1 (Remarks to the Author):

In this manuscript, the authors developed an energy harvesting textile to convert biomechanical energy into electricity. The core technology of the energy harvesting textile is triboelectric nanogenerator (TENG), which is working based on contact electrification and electrostatic induction. The novelty of this manuscript is that the authors fabricated the energy harvesting textile starting from the design of a highly stretchable yarn as a TENG, and they also showed that the highly stretchable yarn has the potential for mass production. They have demonstrated some operation situations and applications of the energy harvesting textile. Overall, my suggestion is that more novelty should be added in order to publish the manuscript in Nature Communications.

1. There are already too many related papers. In those papers, the authors also utilized TENG technology to design their energy harvesting textiles. For example, the paper published in Nature Energy 2016, 1, 16138. They even discussed the weaving types would provide different output performance of the fabricated energy harvesting textile. So the authors in this manuscript should claim the difference to other published papers. For example, providing the output when the energy harvesting textile is under some long-term normal body movements. Like the case of running/walking/riding bicycle 3 hours, how much electricity is generated? Can the generated electricity charge cell phone battery (to what level)?

2. The working mechanism in Figure S7 is not right. The number of positive and negative charges should always be equal at each status.

3. In Figure 3d and the supporting video, I found that even the yarn was not operated, the electricity was still generated (because the number 8 was always shown on the LCD). Was the electricity really generated from the yarn stretch?

3. The authors claimed that the built-in helix structure is advantageous. However, as I know, even the core of the yarn is flat, it would also generate output under the stretch movement. The authors may need to compare and add the data.

4. Will the diameter of the stainless steel yarn affect the output of the yarn? It should be another important factor need to be discussed in the manuscript.

Reviewer #2 (Remarks to the Author):

This paper reports a method to fabricate amphibious triboelectric yarns and textiles. Although wearable triboelectric devices have been already intensively investigated, this paper shows advantages of large-scale manufacturing and its all-weather durability, indicating a considerable advance in practical daily use of wearable triboelectric technology. Also the anomalously better energy-generating performance under water is interesting. However there are some concerns that need to be explained. This paper may be considered for publication after major revisions.

1. In testing the energy-generating performance of the yarns, how is the yarn being stretched as it contains both core and sheath.

2. The paper claimed the triboelectric yarn is amphibious because the sheath silicon tube is waterproof. Are the two ends of the yarns hermetically sealed? If so, how to effectively seal the ends of the core-sheath structure and will this affect the stretching test of the yarns? If not, then will liquid permeates into the tube and spoil the operation of the yarns?

3. The author attributed the anomalously better performance of the energy yarns to the potential-polarization coupling effect. How does it differ from the surface-dielectric polarization effect enabled by ferroelectric materials in a published paper (Achieving ultrahigh triboelectric charge density for efficient energy harvesting. Nature communications, 2017, 8(1): 88.). The author should make a detail comparison between his work and others.

4. The stretchability of the triboelectric yarn is achieved by coupling intrinsic elasticity of rubber and extrinsic elasticity of stainless steel helix. It is notable that the stress/strain transfer in different

materials/structures barely equal. Thus, in addition to contact and separation of two materials as illustrated in Figure 2, friction, which could be induced by asynchronous deformation of sheath and core materials, may also be taken into consideration in discussing in-depth working principle of the SETEYs. The author should clearly explain the above.

5. Figure 3a shows an interesting relationship between the device output and relative dielectric constant of environmental medium. Maybe data curves can help understand the underlying ruling principle of effect.

6. It is suggested to demonstrate a statistical reproducibility of the triboelectric yarn.

7. In line 142, "alternative current" should be "alternating current",

Reviewer #3 (Remarks to the Author):

This paper reported a highly stretchable triboelectric yarn (SETEY) with intrinsically elastic rubber sheath and extrinsically elastic built-in conductive yarn core. Such yarn was fabricated by a modified melt-spinning method, in which stretchable rubber sheath was melt coated on stainless steel yarns, and through compression and stress applied on the sheath, spontaneous of the stainless steel core happened. As such, one can produce these stretchable SETEY in a scalable and continuous way. SETEY shows triboelectric effect both in the air and in liquid environment. Large-area self-powered fabrics were also demonstrated.

The idea of using melt spinning equipment for the one-step and continuous fabrication of stretchable triboelectric core-shell SETEY yarns is novel, but the helical structure of core is not new. The advantages of such method compared to other fabrication methods reported in the literature are not obvious. The expressions on the working mechanism is not clear and experimental results do not always support the discussion. Many important literatures are missing. The novelty and significance of this paper are not suitable for Nat. Comm.

More detailed comments are below:

1. The author stated that scalable manufacture of triboelectric nanogenerator (TENG) textiles were not achieved yet, and TENG had difficulty in weaving. As a matter of fact, continuous manufacture of triboelectric yarns by other spinning approaches and large-area TENG woven/knitted fabrics have been extensively demonstrated in the literature. Waterproof and washable TENG have also been demonstrated. The advantages of such reported approach are not obvious.

For Example:

- a. Lai, Ying-Chih, et al. "Single-thread-based wearable and highly stretchable triboelectric nanogenerators and their applications in cloth-based self-powered human-interactive and biomedical sensing." *Advanced Functional Materials* 27.1 (2017): 1604462.
- b. Yu, Aifang, et al. "Core-Shell-Yarn-Based Triboelectric Nanogenerator Textiles as Power Cloths." *ACS nano* 11.12 (2017): 12764-12771.
- c. Zhao, Zhizhen, et. a., "Machine-washable Textile Triboelectric Nanogenerators for Effective Human Respiratory Monitoring through Loom Weaving of Metallic Yarns." *Advanced Materials* 28 (2016): 10267-10274.
- d. Kwak, Sung Soo, et al. "Fully Stretchable Textile Triboelectric Nanogenerator with Knitted Fabric Structures." *ACS nano* 11.11 (2017): 10733-10741.

2. The mechanism of the SETEY operation is not well presented, and the explanation is not well supported by the data. Lots of parameters that might affect the performance of TENG were mentioned in the manuscript, but they were not clearly described. For example, the author claimed that the overall output performance of SETEY was determined by yarn diameter, structure and their fabrication process, but there was no detailed analysis regarding their relations. In addition, the author expressed much on the parameters of weaving process (e.g., couple moment, torsional moment), but whether they greatly affect the performance of TENG remained unclear.

3. This paper proposed a newly developed melt spinning approach to fabricate the core-shell triboelectric yarn, but there is no detailed characterization of the as-spun yarn. Apart from the schematic illustration, they should provide high-resolution SEM or optical images showing the longitudinal and cross-sectional morphologies of the SETEY yarn at the different stretching and releasing stages.

4. In the last session of the paper, the author incorporated SETEY yarn with polyacrylonitrile/stainless steel yarn to form the woven TENG fabric. Since the author has demonstrated that a single SETEY yarn was functionable as an amphibious wearable power supply, such SETEY yarns in principle should be able to be directly woven or knitted into a piece of highly stretchable TENG fabric. The purpose of incorporating additional yarns is not clearly illustrated.

5. Eventhough the SETEY itself can run under water, the weaved fabrics do not sound a stable TENG underwater. Again, the discussion and the data are quite ambiguous.

Responses to the Reviewers' Comments

Point-by-point responses to the reviewers' comments

We sincerely thank the reviewers for their careful and thorough review, which are indeed very helpful to make the paper more solid and smooth. We have revised our manuscript very carefully in the light of their suggestions and comments.

The following responses have been prepared to address all of the reviewers' comments in a point-by-point fashion. (Comments in black, responses in blue):

For Reviewer #1:

General comment: In this manuscript, the authors developed an energy harvesting textile to convert biomechanical energy into electricity. The core technology of the energy harvesting textile is triboelectric nanogenerator (TENG), which is working based on contact electrification and electrostatic induction. The novelty of this manuscript is that the authors fabricated the energy harvesting textile starting from the design of a highly stretchable yarn as a TENG, and they also showed that the highly stretchable yarn has the potential for mass production. They have demonstrated some operation situations and applications of the energy harvesting textile. Overall, my suggestion is that more novelty should be added in order to publish the manuscript in Nature Communications.

Response: Thank you for your review and we appreciate your feedback. We have carefully revised the manuscript according to your comments. The replies to each of your concern are listed below.

1. There are already too many related papers. In those papers, the authors also utilized TENG technology to design their energy harvesting textiles. For example, the paper published in Nature Energy 2016, 1, 16138. They even discussed the weaving types

would provide different output performance of the fabricated energy harvesting textile. So the authors in this manuscript should claim the difference to other published papers. For example, providing the output when the energy harvesting textile is under some long-term normal body movements. Like the case of running/walking/riding bicycle 3 hours, how much electricity is generated? Can the generated electricity charge cell phone battery (to what level)?

Response: Thank you very much for your insightful review. It is indeed that many researchers have utilized TENG technology to design energy harvesting textiles, therefore according to your suggestions, we would like to emphasize the novelty of our work and provide more clear discussion on the difference of our work to others.

First, we would like to point out the difference between energy harvesting FIBER (or yarn) and TEXTILE. The existing energy harvesting textiles are mostly fabricated by weaving (knitting, or braiding) at least two kinds of fibers (or yarns) that hold opposite electrical properties [References: Adv. Mater. 2016, 28, 10267–10274 (hereinafter referred to as *AM10267*); Adv. Funct. Mater. 2017, 27, 1604462 (hereinafter referred to as *AFM1604462*); Nature Energy 2016, 1, 16138 (hereinafter referred to as *NE16138*)]. These raw fibers themselves do not show TENG behaviors. The design and fabrication of raw fiber or yarn that is individually capable of harvesting energy would be a completely different story, which is the core concept introduced in our submitted work.

The idea of originating energy clothing from fiber/yarn-like TENG has attracted broad attentions. Researches including ourselves have previously shown the possibility to fabricate stretchable and wearable fiber/yarn-like TENG [References: Adv. Energy Mater. 2018, 1801114; Nano Energy 2017, 39, 673–683; Nano Energy 2017, 41, 511–518; Adv. Funct. Mater. 2017, 1604378]. However, the remaining crucial challenges are (1) continuous and scalable manufacture of fiber (or yarn)-TENG, and (2) fabrication of soft (stretchable and flexible) yet mechanically strong ones to meet the requirement of machine weaving in industrial textile and cloth production. In this work,

we report the first ever hundred-meter-length (but not limited to this size) yarn-like TENG that is fabricated continuously. We further confirm that this yarn-like TENG is applicable to industrial post-treatment process such as machine weaving.

More detailed discussions on the superiority in this work to previously reported papers are presented below:

In the *AM10267*, the single-layer textile TENG (t-TENG) was fabricated by plain-weaving 2-ply Cu-PET yarns as warp and PI-Cu-PET yarns as weft. In the *AFM1604462*, the triboelectric threads were fabricated by coating silicone rubber on the stainless-steel threads, then the triboelectric threads can be sewn into a serpentine shape onto the both sides of elastic textile to fabricate the stretchable energy-harvesting textile. First of all, the triboelectric yarns (or threads) reported in these works are not capable of generating electrical energy independently, so such triboelectric yarns (or threads) are not triboelectric nanogenerators in the strict sense. However, our triboelectric yarns can individually capable of harvesting energy. Secondly, these reported triboelectric yarns (or threads) are directly exposed, thereby the output performance of the energy textiles is severely affected by humidity (Figure 2h in the *AFM1604462*). Nevertheless, in our work, the output performance of the triboelectric yarn is completely unaffected by humidity. The good waterproof property of the triboelectric yarn is only one of the highlights in our work. The more important highlight of our work is the discovery of a counterintuitive phenomenon that the triboelectric yarn will generate potential-polarization coupling effect in liquid. Thirdly, the 2-ply Cu-PET yarns and the PI-Cu-PET yarns are obtained by depositing Cu and coating PI on the PET substrate, so their tensile properties are very poor. In our work, the working strain of energy textile can reach 200%. Most important of all, though the single-layer t-TENG in the *AM10267* is woven using an industrial sample weaving loom, the preparation period is roughly as long as 10.5 hours (estimated according to the Experimental section in the *AM10267*). And the triboelectric threads in the *AFM1604462* are manually sewn into the elastic textile to achieve good tensile properties, so this energy textile lacks the potential for industrial applications. In

comparison, our humidity-proof and highly stretchable triboelectric yarn can be spin very fast at a speed of 28 m min^{-1} . It is also worth noting that only commercial and low-cost materials (silicone rubber and stainless steel yarn) are used in our work. Therefore, our continuous production method is more conducive to the application of triboelectric technology in the industrial field.

In the *NEI6138*, the authors prepared an all-solid hybrid power textile so that both light and mechanical energy can be harvested simultaneously. But there still remain many issues to be further addressed. First, similar to the previous two literatures, PTFE electrode and Cu electrode in the *NEI6138* are exposed. In Supplementary Note 3 and Supplementary Figure 22, the authors mentioned that the electrical output of the fabric TENG was weakened by humidity. Second, copper foil is used as an internal electrode in the *NEI6138*. It is well known that the copper foil has a certain flexibility, but its wearability is poor. The movement of the body would cause the copper foil to wrinkle, thereby affecting the current transmission efficiency. Besides, the copper wire was used as a current collector, which also weakens the flexibility of the fabric TENG. Third, the fabrication period of the photoanode exceeds 34 hours, which is obviously not conducive to the scalable manufacture of the hybrid power textile. In comparison, again, we would like to emphasize the alternative solution offered in our work: We have manufactured triboelectric yarns with water resistance and high stretchability by industrial grade melt-spinning method using commercial and low-cost materials. These triboelectric yarns can harvest energy independently and are a fiber/yarn-like TENG.

Finally, according to your kind suggestion, we show that our energy harvesting textiles can charge lithium batteries under some long-term normal body movements (Figure R1a). First, we use the device shown in Figure 4c to simulate long-term normal body movements. The Li-ion batteries (240 mAh, 3 V) are charged by energy textiles, and the battery voltage increment reaches approximately 0.36 V after continuous charging for 5 h (Figure R1b), which indicates that energy textiles have good durability. The pre-charged lithium batteries can further power the wireless monitoring system (Figure R1a and d), so that the body movements can be monitored in real time. As

shown in Figure R1c and Video R1, the energy textile is worn on the elbow of the presenter. The energy textile generates a corresponding electrical signal with the elbow bending, which is then wirelessly transmitted to the cell phone via Bluetooth. As demonstrated in Video R1, the different bending angles of the elbow can generate different electrical signals. Even if the bending angle is very small, there are still electrical signals, which shows excellent sensitivity.

Figure R1. (a) The equivalent circuit of a self-charging wireless monitoring system for real-time monitoring of body movements. (b) The charge curve of the Li-ion battery with the voltages ranged from 2.59 to 2.95 V. (c) A photo of real-time wireless monitoring of elbow bending. (d) A photo of the wireless monitoring system.

Our revision to the manuscript:

We added discussions of the self-charging wireless monitoring system as Note S21: “We show that our energy harvesting textiles can charge lithium batteries under some long-term normal body movements (Figure S41a). First, we use the device shown in Figure 4c to simulate long-term normal body movements. The Li-ion batteries (240 mAh, 3 V) are charged by energy textiles, and the battery voltage increment reaches approximately 0.36 V after continuous charging for 5 h (Figure S41b), which indicates

that energy textiles have good durability. The pre-charged lithium batteries can further power the wireless monitoring system (Figure S41a and d), so that the body movements can be monitored in real time. As shown in Figure S41c and Video S7, the energy textile is worn on the elbow of the presenter. The energy textile generates a corresponding electrical signal with the elbow bending, which is then wirelessly transmitted to the cell phone via Bluetooth. As demonstrated in Video S7, the different bending angles of the elbow can generate different electrical signals. Even if the bending angle is very small, there are still electrical signals, which shows excellent sensitivity.” We also added “Finally, as a real example, we show that our energy textiles are integrated into a wireless monitoring system for real-time monitoring of body movements (Figure S41, Video S7 and Note S21). The energy harvesting textiles can generate varying electrical signals with the elbow bending, which are then wirelessly transmitted to the cell phone via Bluetooth. Meanwhile, the energy textiles show potential to charge Li-ion batteries of the wireless monitoring system. As a wearable self-powered active kinematic sensor or power supply, the scalable-manufactured energy yarns and textiles are readily working, while there is still space for further improvement as for the devices electrical output.” in the revised manuscript (*part Electrical output performance of the e-textiles*). Figure R1 and Video R1 were added as Figure S41 and Video S7, respectively. We also added “Video S7. This video shows a self-charging wireless monitoring system for real-time monitoring of body movements.” in the Supplementary Information (*part Caption for Supporting Videos*).

Corresponding changes have been marked in red in the revised manuscript.

2. *The working mechanism in Figure S7 is not right. The number of positive and negative charges should always be equal at each status.*

Response: Thank you very much for your comment on the working mechanism. As the reviewer said, the number of positive and negative charges of the entire system should always be equal at each status, which is also applicable to our single-electrode mode.

For instance, in Figure R2 [Reference: "Triboelectric Nanogenerator: Single-Electrode Mode." *Triboelectric Nanogenerators*. Springer International Publishing, 2016, 91-107.], the number of negative charges at the fluorinated ethylene propylene (FEP) layer is equal to positive charges at aluminum layer (primary electrode) and copper layer (reference electrode).

However, the number of charges seems to change if the ground is considered as a reference electrode, as illustrated in Figures R3 [Reference: *ACS Nano*, 2013, 7(8): 7342-7351.] and R4 [Reference: *Adv. Mater.* 2013, 25(45): 6594-6601.]. This is because that the ground is a conductor, and it is accustomed to use the symbol to represent the ground when interpreting the working mechanism of the TENG. But in fact, a part of the positive charges is hidden in the symbol of the ground and is not presented in Figure S9.

We apologize for the imperfect explanation of the working mechanism. We have provided additional instructions in the revised manuscript (Note S7) as follows:

“The SETEY is a single-electrode triboelectric nanogenerator. In order to better understand the working mechanism of SETEY, we present its full working cycle through the front view (Figure S9a1-d1) and the sectional drawing (Figure S9a2-d2), respectively. It is notable that the number of positive and negative charges is always equal in the front view and the side view. However, in the schematic diagram, partial charges are hidden due to differences in viewing angles and are not presented. In the entire system, all negative charges are located on the internal surface of the silicone rubber tube, and all positive charges are located on the stainless steel yarn and the ground electrode.”

In addition, we find that we made a serious mistake in Figure S9 of the previously submitted manuscript, and we reversed the labels of positive and negative charge. We apologize for this major mistake.

Corresponding changes have been marked in red in the revised manuscript.

Figure R2. Working mechanism of the conductor-to-dielectric single-electrode TENG based on vertical contact separation. [Reference: "Triboelectric Nanogenerator: Single-Electrode Mode." Triboelectric Nanogenerators. Springer International Publishing, 2016, 91-107.]

Figure R3. Sketches that illustrate the electricity generation process in a full cycle. [Reference: ACS Nano, 2013, 7(8): 7342-7351.]

Figure R4. The sketches that illustrate the electricity generation process in a full cycle of the TENG. [Reference: Adv. Mater., 2013, 25(45): 6594-6601.]

Figure S9. Schematic diagram of the working mechanism of SETEY's (a₁-d₁) front view and (a₂-d₂) side view, respectively.

3. In Figure 3d and the supporting video, I found that even the yarn was not operated, the electricity was still generated (because the number 8 was always shown on the LCD). Was the electricity really generated from the yarn stretch?

Response: Thank you your careful review. To explain this phenomenon, we would like to introduce the principle of liquid crystal display (LCD) first: It is based on the fact that the transmittance of a liquid crystal varies with the magnitude of the applied voltage. When the accumulated residual charges cause the applied voltage between the two electrodes to exceed the threshold voltage, the liquid crystal molecules will turn, thereby the transmittance of the liquid crystal will also change [References: Japanese Journal of Applied Physics, 2009, 48(5R): 055002; Japanese journal of applied physics, 2007, 46(9L): L796; Applied physics letters, 1998, 73(20): 2881-2883].

In Figure 3d and the supporting video, a LCD that has been operated for multiple

times was used in this demonstration experiment. Due to that the residual charges on the LCD, a very small signal, i.e., a faint “8” pattern is visible. If we look closely, it is clear that the pattern becomes clear (or the signal becomes strong) as the yarn is stretched, indicating that the yarn stretch does produce electricity.

To further verify our conclusion, we used a new LCD that has negligible residual charges to demonstrate our researches during revision. As can be seen in the revised Video S4, the “8” pattern on the LCD is determined by the electric field generated from the yarn stretching.

We have made corresponding revisions to Figure 3d and Video S4 in the revised manuscript.

4. The authors claimed that the built-in helix structure is advantageous. However, as I know, even the core of the yarn is flat, it would also generate output under the stretch movement. The authors may need to compare and add the data.

Response: Thanks a lot for your professional comment. We may need to point out the difference in underlying mechanism between our built-in helix structure and flat-yarn structure first. In a flat-yarn structure, the core and shell materials should be both stretchable and at least one of them should be conductive. There raises concerns: (1) the resistance change of the conductive material under stretch movement will interfere (the uniformity and readability of) electrical signals, (2) in a flat structure, there is a lack of sufficient transitional state between the two states of full contact and complete separation, resulting in its electrical strain sensitivity being low, and (3) under stretch movement, the flat core and shell deform very similarly, i.e., elongate at almost same tensile strain, therefore few friction can be expected.

In comparison, our built-in helix structure addresses the resistance-variation issue (Figure R6b). Besides, the built-in helix structure achieves the effective transition between full contact and complete separation by the change of the helix angle (i.e., the degree of the helix) (Figure R5). We summarize key performances of two different

structures in Table R1. The advantage of our results is clear.

In addition, based on the same materials and similar fabrication method reported in this work, we fabricate a core-shell yarn-like TENG with a flat core yarn. Its structure and mechanism are illustrated in Figure R6a. As shown in Figure R6c, it is obvious that the built-in helix structure has a larger and more stable output voltage than the flat structure. The reasons are: (1) the silicone rubber tube has a larger contact area with the stainless steel yarn in the built-in helix structure, which causes it to have a larger output voltage than the flat structure. (2) the built-in helix structure obtains a fixed number of helices by pre-stretching, thereby maintaining a stable contact-separation area; however, the contact area of the flat structure after each stretching may be somewhat deviated, resulting in that the output is not stable.

Table R1. The comparison of our work with recently reported related works, where R_0 is the initial resistance, ΔR is the variation of the resistance, $\Delta V/V_0$ is the normalized voltage change, and $\Delta \varepsilon$ is the variation of the tensile strain.

	Built-in helix	Flat structure			
	This work	Adv. Energy Mater. 2018, 1801114	Nano Energy 2017, 41, 511–518	Adv. Funct. Mater. 2017, 1604378	Scientific reports, 2016, 6: 35153.
Work strain	200%	100%	60%	70%	50%
Relative resistance variation at strain of 50% ($\Delta R/R_0$)	0.0013	--	0.2	--	0.28
Gauge factor	38.2	10	10	11	2.1

$(GF = \frac{\Delta V/V_0}{\Delta \varepsilon})$					
--	--	--	--	--	--

Figure R5. The longitudinal morphologies of the SETEY under different strains during stretching.

Figure R6. (a) The structure and mechanism of a core-shell yarn-like TENG with a flat core yarn. (i) Schematic description of the core-shell yarn-like TENG with the silicone rubber tube stretched outward. (ii) Schematic diagram of the working principle of the

core-shell yarn-like TENG under the stretching state. (b) The conductivity variation of the core electrode of the SETEY under stretching test. (c) The output voltage of built-in helix structure and flat structure at a fixed tensile strain of 100%.

Our revision to the manuscript:

We added discussions of the flat-yarn structure as Note S10:

“We may need to point out the difference in underlying mechanism between our built-in helix structure and flat-yarn structure first. In a flat-yarn structure, the core and shell materials should be both stretchable and at least one of them should be conductive. There raises concerns: (1) the resistance change of the conductive material under stretch movement will interfere (the uniformity and readability of) electrical signals, (2) in a flat structure, there is a lack of sufficient transitional state between the two states of full contact and complete separation, resulting in its electrical strain sensitivity being low, and (3) under stretch movement, the flat core and shell deform very similarly, i.e., elongate at almost same tensile strain, therefore few friction can be expected.

In comparison, our built-in helix structure addresses the resistance-variation issue (Figure S23b). Besides, the built-in helix structure achieves the effective transition between full contact and complete separation by the change of the helix angle (i.e., the degree of the helix) (Figure S14a).

In addition, based on the same materials and similar fabrication method reported in this work, we fabricate a core-shell yarn-like TENG with a flat core yarn. Its structure and mechanism are illustrated in Figure S23a. As shown in Figure S23c, it is obvious that the built-in helix structure has a larger and more stable output voltage than the flat structure. The reasons are: (1) the silicone rubber tube has a larger contact area with the stainless steel yarn in the built-in helix structure, which causes it to have a larger output voltage than the flat structure. (2) the built-in helix structure obtains a fixed number of helices by pre-stretching, thereby maintaining a stable contact-separation area; however, the contact area of the flat structure after each stretching may be somewhat deviated, resulting in that the output is not stable.”

We also added “This working mechanism of the built-in helix SETEY causes it to exhibit completely different features than the common flat structure. We compared the difference between the built-in helix structure and the flat structure and found that the output performance of the former is larger and more stable (Note S10, Figure S14a, and Figure S23).” in the revised manuscript (*part Working principle of the SETEYs*). Figure R5 and Figure R6 were added as Figure S14a and Figure S23, respectively.

Corresponding changes have been marked in red in the revised manuscript.

5. Will the diameter of the stainless steel yarn affect the output of the yarn? It should be another important factor need to be discussed in the manuscript.

Response: Thanks a lot for your helpful suggestion. In this revision, we discussed the effect of the diameter of the stainless steel yarn on the output of the yarn, as shown in Figure R7. We selected a silicone rubber tube with an inner diameter of 0.8 mm as the friction sheath layer. The overall output voltage of SETEY decreases as the diameter of the stainless steel yarn increases. However, when the diameter of the stainless steel yarn is less than 0.45 mm, the output performance changes very slightly. When the diameter of the stainless steel yarn exceeds 0.45 mm, the output performance drops dramatically. We have also discussed on the effect of the inner diameter of sheath tube, as shown in Figure S10b. It can be found that the output voltage of SETEY first increases with increasing inner diameter, then decreases when the inner diameter exceeds 1.5 mm.

Above phenomena are attributed to the variation of contact area and separation distance between core and sheath materials. It is known that the performance of single-electrode mode TENG is greatly affected by the separation distance when the separation distance is small. However, this effect will gradually decrease as the separation distance increases [References: Adv. Funct. Mater. 2014, 24, 3332–3340]. Here, according to Equation 11 in Note S9, Figures S10b, d, e and g, we know that the performance is mainly affected by the separation distance when the separation distance is less than 0.305 mm ($D_0 = 1.5$ mm and $D_y = 0.45$ mm). When the separation distance is greater

than 0.305 mm, the influence of the separation distance is weakened, and the performance is mainly affected by the contact area. Further, when the diameter of the stainless steel yarn is less than 0.45 mm, the change in the output performance is not obvious. However, the 0.45-mm stainless steel yarn has greater breaking strength at this time (Figure R8), so we chose a 0.45-mm stainless steel yarn as the core yarn of SETEY.

It is notable that when the silicone rubber tube is stretched outward, its length increases and the inner diameter becomes smaller, as schematically shown in Figure R9. Since the internal volume of the silicone rubber tube is always constant before and after stretching, the inner diameter of the silicone rubber tube has the following expression:

$$\pi\left(\frac{D_0}{2}\right)^2 L_0 = \pi\left(\frac{D_s}{2}\right)^2 L_s \quad (1)$$

$$D_s = D_0 \sqrt{\frac{L_0}{L_s}} \quad (2)$$

where D_0 is the inner diameter of the silicone rubber tube in the original state, L_0 is the length of the silicone rubber tube in original state, D_s is the inner diameter of the silicone rubber tube in the pre-stretching state, and L_s is the length of the sheath fiber tube in pre-stretching state. Furthermore, we find that the inner diameter of the silicone rubber tube and its pre-strain have the following relationship:

$$\varepsilon = \frac{L_s - L_0}{L_0} \quad (3)$$

$$D_s = D_0 \sqrt{\frac{1}{1+\varepsilon}} \quad (4)$$

where ε is pre-strain of the silicone rubber tube. Note that regardless of how the inner diameter of the silicone rubber tube changes, it is always larger than the diameter of the stainless steel yarn during the stretching of the silicone rubber tube. Therefore, the diameter of the stainless steel yarn has the following relationship with the inner diameter and pre-strain of the silicone rubber tube:

$$D_y < D_0 \sqrt{\frac{1}{1+\varepsilon}} \quad (5)$$

where D_y is diameter of the stainless steel yarn. Inequality (5) can be taken as a consideration for selecting a stainless steel yarn of a suitable diameter.

Figure R7. Voltage versus the diameter of the stainless steel yarn curves measured under 100% tensile strain.

Figure R8. Tensile stress versus strain curve of the stainless steel yarns of different diameters (diameter: 0.35mm, 0.4mm, 0.45mm).

Figure R9. Schematic depiction of the silicone rubber tube being stretched outward.

Our revision to the manuscript:

We added the above discussions to Note S9. Figures R7, R8 and R9 were added as Figures S10e, S10h and S10a, respectively.

Corresponding changes have been marked in red in the revised manuscript.

Thank you again for your great help & Merry Christmas.

For Reviewer #2:

General comment: *This paper reports a method to fabricate amphibious triboelectric yarns and textiles. Although wearable triboelectric devices have been already intensively investigated, this paper shows advantages of large-scale manufacturing and its all-weather durability, indicating a considerable advance in practical daily use of wearable triboelectric technology. Also the anomalously better energy-generating performance under water is interesting. However there are some concerns that need to be explained. This paper may be considered for publication after major revisions.*

Response: Thank you for the positive feedback and we will try our best to address the questions raised.

1. In testing the energy-generating performance of the yarns, how is the yarn being stretched as it contains both core and sheath.

Response: Thanks for your valuable comments. On the one hand, the sheath rubber tube itself has extremely excellent tensile properties, which exhibits an elastic strain of 200% and a maximum tensile strain of 633% (Figure S3 and S4). On the other hand, as demonstrated in Figure R1, the core yarn presents helical structure inside the sheath rubber tube. When the sheath rubber tube is stretched, the number of the helix remains constant and the helix angle (i.e., the degree of the helix) is reduced, which causes the core yarn to have extrinsic elasticity. Therefore, the stretchability of the triboelectric yarn is achieved by coupling intrinsic elasticity of sheath rubber tube and extrinsic elasticity of core yarn.

Figure R1. The longitudinal morphologies of SETEY under different strains during stretching.

Our revision to the manuscript:

Figure R1 was added as Figure S14a. Corresponding changes have been marked in red in the revised manuscript.

2. *The paper claimed the triboelectric yarn is amphibious because the sheath silicon tube is waterproof. Are the two ends of the yarns hermetically sealed? If so, how to effectively seal the ends of the core-sheath structure and will this affect the stretching test of the yarns? If not, then will liquid permeates into the tube and spoil the operation of the yarns?*

Response: Thanks a lot for your professional comments. It is indeed that the two ends of the triboelectric yarn are hermetically sealed. As shown in Figure R2, we used a commercial two-component epoxy resin AB adhesive to seal the two ends of the triboelectric yarn. The AB adhesive has the advantages of good adhesion, good flexibility, good transparency, water resistance, etc., and it can be cured quickly in 5 minutes at room temperature. Therefore, the AB adhesive can prevent liquid from permeating into the sheath silicon tube and does not affect the appearance of the triboelectric yarn. In addition, since the AB adhesive is only at two ends of the

triboelectric yarn, and the two ends of the triboelectric yarn are fixed during the stretching test, the AB adhesive does not affect the stretching of the triboelectric yarn.

Figure R2. (a) and (b) are respectively a front view and a side view of SETEY with good sealing properties.

3. The author attributed the anomalously better performance of the energy yarns to the potential-polarization coupling effect. How does it differ from the surface-dielectric polarization effect enabled by ferroelectric materials in a published paper (*Achieving ultrahigh triboelectric charge density for efficient energy harvesting. Nature communications, 2017, 8(1): 88.*). The author should make a detail comparison between his work and others.

Response: Thank you very much for your insightful review and providing related reference. We feel sorry that we did not notice this paper. The reference mentioned by you is a nice work and provides enlightening results and we have cited it as Ref. 44 in the revised manuscript.

In fact, to some extent, our potential-polarization coupling effect is similar to the surface-dielectric polarization effect between triboelectrification surface and ferroelectric material in this reference. In our work and this reference, both water and BT (barium titanate) have residual polarization due to dielectric relaxation, which will enhance the polarization of the triboelectrification surface in subsequent contact

electrification processes, thereby enhancing the amount of triboelectric charges.

It is worth to note that water exhibits a distinct layered structure near surface of the silicone rubber tube (Figure 3c). The dielectric constant K_i of interfacial water is only ~ 2 due to the presence of a dead layer with vanishingly small polarization. The dead layer extends two to three molecular diameters ($h_i = 1.5$ to 2 nm) away from the surface of the silicone rubber tube, which are difficult to reorient by applying an electric field. Therefore, the surface potential from triboelectrification only interacts with bulk water ($K_b \approx 80$), and the extremely thin interface water has no effect on the entire interaction process. However, the entire built-in ferroelectric layer interacts with the polarization of the triboelectrification surface in the reference.

The related presentation has been modified (marked in red) in the revised manuscript: “To some extent, the potential-polarization coupling effect is similar to the surface-dielectric polarization effect between triboelectrification surface and ferroelectric material [44]”

4. The stretchability of the triboelectric yarn is achieved by coupling intrinsic elasticity of rubber and extrinsic elasticity of stainless steel helix. It is notable that the stress/strain transfer in different materials/structures barely equal. Thus, in addition to contact and separation of two materials as illustrated in Figure 2, friction, which could be induced by asynchronous deformation of sheath and core materials, may also be taken into consideration in discussing in-depth working principle of the SETEYs. The author should clearly explain the above.

Response: Thanks for your careful comments. Indeed, as the reviewer said, the stress/strain transfer in different materials/structures barely equal under normal conditions. However, there are some abnormal phenomena in our work due to the special material selection and structural design.

On the one hand, as shown in Figure S3 and Video S3, the stretching of the silicone rubber tube is a linear elastic deformation obeying Hooke's law when the tensile strain

is less than 200%, so the stress/strain on the silicone rubber tube is uniformly transferred. On the other hand, the deformation of the stainless steel helix with extrinsic elasticity is simultaneously affected by the two sealed ends of the triboelectric yarn and the inner wall of the silicone rubber tube. It is worth noting that the stretch length of each point of a helical structure (such as a spring) obeying Hooke's law follows a certain relationship during stretching. As shown in Figure R3, it is assumed that the left end of the helical structure 1 is fixed, and a force is applied on its right end to stretch forward to obtain the helical structure 2. There is the following relationship between the helical structure 1 and the helical structure 2:

$$dL_m = \frac{L_m}{L_n} dL_n \quad (1)$$

where L_n is the distance from point n in the helical structure 1 to the left end of the helical structure 1 (i.e., the length of the helical structure 1), and L_m is the distance from point m in the helical structure 1 to the left end of the helical structure 1. dL_n and dL_m are stretching distance of points n and m, respectively, when helical structure 1 is stretched to helical structure 2. In Figure R4 and Video R1, the marks a and b on the triboelectric yarn are stretched to marks A and B, respectively, after 2.792 s. In Figure R4a, the distance from mark a to the left end of the triboelectric yarn is approximately half the distance from the mark b to the left end. From the scale in the Figure R4b, it can be seen that the stretching distance of b is exactly twice the stretching distance of a. Therefore, the stainless steel helix also obeys Hooke's law during the operation of the triboelectric yarn. In addition, as shown in Figure R5 and Video R1, the stainless steel yarn and the silicone rubber tube are marked with y and t, respectively. The stainless steel yarn and the silicone rubber tube are always synchronously deformed during triboelectric yarn stretching (y1 to y4, t1 to t4). And Figure S4 further confirms that the triboelectric yarn has good mechanical cycle stability. In summary, our triboelectric yarns do not undergo asynchronous deformation during stretching.

Furthermore, when the triboelectric yarn is woven from one dimension into a two-dimensional fabric, there will be more stress locations at two ends of the fabric during stretching, which is also conducive to avoiding asynchronous deformation.

Figure R3. Schematic diagram of the helical structure from the original state to the stretched state. It is worth noting that the helical structure 1 and the helical structure 2 are different states of the same object.

Figure R4. (a) An optical photograph of a triboelectric yarn at 8.875 s, where the marks a and b are the two helices at $1/4$ and $1/2$ of the triboelectric yarn, respectively. (b) An optical photograph of a triboelectric yarn at 10.667 s, wherein the marks A and B are positions where the marks a and b are stretched, respectively, and the black line is used as a scale to indicate the distance before and after stretching.

Figure R5. During stretching, the triboelectric yarns were photographed at (a) 22 s, (b) 23 s, (c) 24 s, and (d) 25 s, respectively, where stainless steel yarn and silicone rubber tube were marked with y and t, respectively.

Our revision to the manuscript:

We added the above discussions as Note S5. We also added “Besides, the stainless steel yarn and the silicone rubber tube are always synchronously deformed during the SETEY stretching (Note S5, Figures S6 and S7).” in the revised manuscript (*part Continuous and scalable manufacture of the SETEYs*). Figures R3 and R4 were added as Figures S6. Figure R5 was added as Figure S7.

Corresponding changes have been marked in red in the revised manuscript.

5. *Figure 3a shows an interesting relationship between the device output and relative dielectric constant of environmental medium. Maybe data curves can help understand the underlying ruling principle of effect.*

Response: We thank for the reviewer’s constructive suggestions. As shown in the Figure R6, there is a nonlinear relationship between the relative dielectric constant and the voltage by fitting, which can be expressed by the following formula:

$$V = \frac{abK^{1-c}}{1+bK^{1-c}} \quad (2)$$

where K is the relative dielectric constant of the liquid and V is the output voltages of the triboelectric yarn. a , b and c are the equilibrium constants under normal temperature and atmospheric pressure, where they are 14, 0.385 and 0.17, respectively. As the relative dielectric constant of the liquid increases, the output voltage of the triboelectric yarn begins to increase rapidly, and then the growth trend slows down, that is, the slope of the fitting curve gradually decreases. The cause of this phenomenon may be complex, and some hidden factors may play particular roles in output changes. We believe that there are two most likely factors. First, the output voltage is positively correlated with the polarity of the liquid molecules as the polarity of the liquid molecules changes. The other is that interfacial liquid and bulk liquid may also exist similar to water. The thickness and nature of the interfacial liquids are inconsistent in different liquids, which may be the main factor leading to the change in the slope of the fitting curve. The exploration of the liquid interface is an extremely complex and far-reaching study, and we will further study this factor in more detail in future work.

Figure R6. Relationship between the relative dielectric constant of the liquid and the

output voltages of the triboelectric yarn.

Our revision to the manuscript:

We added the above discussions as Note S13:

“As shown in the Figure S26, there is a nonlinear relationship between the relative dielectric constant and the voltage by fitting, which can be expressed by the following formula:

$$V = \frac{abK^{1-c}}{1+bK^{1-c}} \quad (15)$$

where K is the relative dielectric constant of the liquid and V is the output voltages of the triboelectric yarn. a , b and c are the equilibrium constants under normal temperature and atmospheric pressure, where they are 14, 0.385 and 0.17, respectively. As the relative dielectric constant of the liquid increases, the output voltage of the triboelectric yarn begins to increase rapidly, and then the growth trend slows down, that is, the slope of the fitting curve gradually decreases. The cause of this phenomenon may be complex, and some hidden factors may play particular roles in output changes. We believe that there are two most likely factors. First, the output voltage is positively correlated with the polarity of the liquid molecules as the polarity of the liquid molecules changes. The other is that interfacial liquid and bulk liquid may also exist similar to water. The thickness and nature of the interfacial liquids are inconsistent in different liquids, which may be the main factor leading to the change in the slope of the fitting curve.”

We also added “Besides, Figure 3a also shows an interesting relationship between the device output and relative dielectric constant of environmental medium. The growth trend of the output voltage of the SETEY gradually decreases as the relative dielectric constant increases (Note S13 and Figure S26).” in the revised manuscript (*part Amphibious energy yarns working in liquid*). Figure R6 was added as Figure S26.

Corresponding changes have been marked in red in the revised manuscript.

6. *It is suggested to demonstrate a statistical reproducibility of the triboelectric yarn.*

Response: Thanks for your valuable suggestions. On the one hand, as shown in Figure S18, a single triboelectric yarn maintains good stability and reliability after 2000 tensile tests. On the other hand, we present another two samples (sample 2 was fabricated on September 20 and 3 was fabricated on September 27, while sample 1 presented in original manuscript was fabricated on January 14) in the revised manuscript. The output voltages of the two samples under stretching mode (strains of 10%, 40%, 70% and 100%) were collected and compared with the data from the manuscript (Figure S13, denoted as sample 1), as seen in Figure R7 below. From the comparison, the three samples exhibited similar output voltage at different stretching strains, although with a slight discrepancy. We think this slight discrepancy may be normal production error and test error, which still indicates that our triboelectric yarns have good reproducibility. We will further optimize assembly techniques and test methods to eliminate the discrepancy.

Figure R7. The output voltage comparison of three samples 1, 2 and 3. (a) Output voltages under stretching strain of 10%; (b) Output voltages under stretching strain of 40%; (c) Output voltages under stretching strain of 70%; (d) Output voltages under stretching strain of 100%.

Our revision to the manuscript:

We added the above discussions to Note S9:

“We also present another two samples (sample 2 was fabricated on September 20 and 3 was fabricated on September 27, while sample 1 presented in original manuscript was fabricated on January 14) in the revised manuscript. The output voltages of the two samples under stretching mode (strains of 10%, 40%, 70% and 100%) were collected and compared with the data from the manuscript (Figure S13, denoted as sample 1), as seen in Figure S19 below. From the comparison, the three samples exhibited similar output voltage at different stretching strains, although with a slight discrepancy. We think this slight discrepancy may be normal production error and test error, which still indicates that our triboelectric yarns have good reproducibility.”

Figure R7 was added as Figure S19. Corresponding changes have been marked in red in the revised manuscript.

7. In line 142, “*alternative current*” should be “*alternating current*”.

Response: Thanks for your kind reminder. The related problems have been modified (marked in red) in the revised manuscript.

Thank you again for your great help & Merry Christmas.

For Reviewer #3:

General comment: *This paper reported a highly stretchable triboelectric yarn (SETEY) with intrinsically elastic rubber sheath and extrinsically elastic built-in conductive yarn core. Such yarn was fabricated by a modified melt-spinning method, in which stretchable rubber sheath was melt coated on stainless steel yarns, and through compression and stress applied on the sheath, spontaneous of the stainless steel core happened. As such, one can produce these stretchable SETEY in a scalable and continuous way. SETEY shows triboelectric effect both in the air and in liquid environment. Large-area self-powered fabrics were also demonstrated.*

The idea of using melt spinning equipment for the one-step and continuous fabrication of stretchable triboelectric core-shell SETEY yarns is novel, but the helical structure of core is not new. The advantages of such method compared to other fabrication methods reported in the literature are not obvious. The expressions on the working mechanism is not clear and experimental results do not always support the discussion. Many important literatures are missing. The novelty and significance of this paper are not suitable for Nat. Comm.

Response: We are very grateful to the reviewer for confirming the novelty of our scalable SETEYs fabrication. We would like to emphasize that continuous and scalable manufacture is the most important highlight of this work. Indeed, as the reviewer has said, the helical structure has been reported in other literatures, but it should be noted that our built-in helix is essentially different from the previously reported helical structure. The well-known helix can be divided into two types: one is a self-helical structure similar to a spring structure; the other is an outer helix structure covered on an elastic polymer substrate. The built-in helix reported in this work has completely different features compared with above two mainstream structures: (1) The self-helical structure is generally formed by excessive twisting, and the outer helix structure is generally prepared by coiling. However, the built-in helix is formed automatically by the inherent friction between the sheath and core materials. In Table R1, both the self-

helical structure and the outer helix require two or more steps to prepare the device, while our built-helix can be fabricated into a device by one-step process. (2) Compared with self-helical structure and outer helix structure, our built-in helix has a greater tensile strain (work strain $\approx 200\%$, maximum tensile strain $\approx 600\%$) (Table R1). (3) The built-in helix can be fully stretched straight due to the special forming method, which is especially important in our overall design, but the self-helical structure and the outer helix structure are difficult to do this.

We also thank you very much for your comment on the working mechanism and experimental results. In Note S7, we have further modified (marked in red) the working mechanism and highlighted in red as follows: “The SETEY is a single-electrode triboelectric nanogenerator. In order to better understand the working mechanism of SETEY, we present its full working cycle through the front view (Figure S9a₁-d₁) and the side view (Figure S9a₂-d₂), respectively. It is notable that the number of positive and negative charges is always equal in the front view and the sectional drawing. However, in the schematic diagram, partial charges are hidden due to differences in viewing angles and are not presented. In the entire system, all negative charges are located on the internal surface of the silicone rubber tube, and all positive charges are located on the stainless steel yarn and the ground electrode.” Besides, in the following detailed responses, we carefully revised the deficiencies of the working mechanism and experimental results.

In order to better improve our manuscript, we have further studied additional important literatures. Useful literatures have been cited in the revised manuscript. We have also commented some important literatures in the following detailed responses.

At last, we would like to emphasize more clearly the novelty and significance of our work:

a) By using a newly developed melt-spinning method, we realize continuous and scalable manufacture of triboelectric yarn. A first ever hundred-meter-length (but not limited to this size) triboelectric yarn is demonstrated. It is worth noting that our triboelectric yarn itself is a triboelectric nanogenerator that can generate electricity

independently.

b) The performance of the energy yarn rises from unique structure designs based on intrinsically elastic silicone rubber tubes and extrinsically helix-like built-in stainless steel yarns. The formation of the built-in helix is a self-organized process. Besides, only commercial and low-cost materials (silicone rubber and stainless steel yarn) are used. Therefore, it lays the foundation of the popularization of the manufacture method.

c) According to the potential well model, the reasonable assumptions about the charge transfer mechanism of the certain energy conversion system is proposed for the first time.

d) The water induced short circuit problem, a major concern for most electronic devices, is solved in the energy yarn and its textiles.

e) In particular, studying its energy conversion mechanism, a theory about the potential-polarization coupling effect occurred on the surface of the energy yarn is demonstrated for the first time. It helps to understand the interesting behavior of the energy yarn in liquid.

Overall, we develop and apply a continuous spinning technology for scalable-manufacture of amphibious energy yarns, which integrates only commercial and low-cost materials. Weaving this energy yarn with other commercial yarns, we develop a mechanically stable, highly stretchable, and large-scale amphibious energy textile.

In addition, the scalable manufacture of materials [References: *Nature Materials* 2014, 13, 624 (hereinafter referred to as *NM624*); *Science* 2017, 355, 1062–1066] and devices [References: *Science Advances* 2018, 4, eaap9104; *Nature Nanotechnology* 2016, 11(12): 1010] in many fields has attracted great attention from researchers. In particular, there is an urgent need for low-cost and large-scale energy devices in the field of environmental energy enrichment [References: *Nature Reviews Materials*, 2018, 3(4): 18017; *Nature Energy*, 2017, 2(5): 17038]. However, the scalable manufacture of biomechanical energy harvesting textiles that convert human kinetic energy into electrical energy are still rarely reported. In the first sentence of *NM624*, the author pointed out: “To progress from the laboratory to commercial applications, it will be

necessary to develop industrially scalable methods”. Our work has addressed some of the issues from laboratory to commercial applications. Therefore, in our opinion, the novelty and significance of our work are suitable for Nature Communications.

Table R1. The working strain and the number of preparation steps of the helical structure in different works.

	Built-in helix	Self-helical structure		Outer helix structure	
	This work	Adv. Energy Mater. 2016, 1600976	Science 2017, 357, 773–778	Adv. Funct. Mater. 2015, 25, 1798–1803	Adv. Funct. Mater. 2017, 1604378
Work strain	200%	50%	30%	25%	70%
Preparation step	One step	Three steps	Three steps	Two steps	Three steps

1. The author stated that scalable manufacture of triboelectric nanogenerator (TENG) textiles were not achieved yet, and TENG had difficulty in weaving. As a matter of fact, continuous manufacture of triboelectric yarns by other spinning approaches and large-area TENG woven/knitted fabrics have been extensively demonstrated in the literature. Waterproof and washable TENG have also been demonstrated. The advantages of such reported approach are not obvious.

For Example:

a. Lai, Ying-Chih, et al. "Single-thread-based wearable and highly stretchable triboelectric nanogenerators and their applications in cloth-based self-powered human-interactive and biomedical sensing." *Advanced Functional Materials* 27.1 (2017): 1604462.

b. Yu, Aifang, et al. "Core-Shell-Yarn-Based Triboelectric Nanogenerator Textiles as

Power Cloths." ACS nano 11.12 (2017): 12764-12771.

c. Zhao, Zhizhen, et. a., "Machine-washable Textile Triboelectric Nanogenerators for Effective Human Respiratory Monitoring through Loom Weaving of Metallic Yarns." Advanced Materials 28 (2016): 10267-10274.

d. Kwak, Sung Soo, et al. "Fully Stretchable Textile Triboelectric Nanogenerator with Knitted Fabric Structures." ACS nano 11.11 (2017): 10733-10741.

Response: Thanks for your comment. It is indeed that many researchers have utilized TENG technology to design TENG fabrics. Even so, our triboelectric yarns still have some commendable advantages. Therefore, we would like to emphasize the novelty of our work and provide more clear discussion on the difference of our work to others.

(1) In Lai, Ying-Chih, et al.'s paper (Advanced Functional Materials 27.1 (2017): 1604462, hereinafter referred to as *AFM1604462*), a new type of triboelectric thread and its uses in elastically textile-based energy harvesting and sensing have been demonstrated. In their work, the triboelectric threads were fabricated by coating silicone rubber on the stainless-steel threads, then the triboelectric threads can be sewn into a serpentine shape onto the both sides of elastic textile to fabricate the stretchable energy-harvesting textile.

It is worth noting that although the authors stated in the *AFM1604462* that the triboelectric thread is a triboelectric nanogenerator, in fact, the independent triboelectric thread along does not produce electricity, and it requires movement relative to the skin to generate electricity. In other words, it is necessary to combine triboelectric threads and skin to demonstrate so-called triboelectric nanogenerator. Besides, electrons transfer between the triboelectric threads and the skin, thereby the authors stated in the *AFM1604462* that performance of the energy-harvesting textile is severely affected by humidity (Figure 2h in the *AFM1604462*). In comparison, in our work, the output performance of the triboelectric yarn is completely unaffected by humidity. More importantly, we reveal a counterintuitive phenomenon that the triboelectric yarn will generate potential-polarization coupling effect in liquid.

In addition, the triboelectric threads in the *AFM1604462* are manually sewn into the elastic textile to achieve good tensile properties, so this energy textile lacks the potential for industrial applications. In comparison, our work demonstrates machine spinning and weaving processes, which can be used for industrial production, and is another major significance of our work.

(2) In Yu, Aifang, et al.'s paper (ACS Nano 11.12 (2017): 12764-12771, hereinafter referred to as *ACS12764*), TENG textiles woven from core-shell yarns were successfully fabricated to scavenge various types of human motion energies as energy-harvesting cloths. The core-shell yarns were woven from core conductive fibers and sheath dielectric fibers, wherein the core conductive fibers served as electrodes and the shell dielectric fibers served as electrification layers.

Similar to the triboelectric thread reported in the *AFM1604462*, the individual core-shell yarn reported in the *ACS12764* is not a triboelectric nanogenerator, which needs to be woven into the TENG textile to generate electricity. The output performance of the exposed core-shell yarns will be severely weakened by humidity during operation too, which will greatly limit the application environment and range of the yarn.

In addition, the tensile performance of the core-shell yarn in the *ACS12764* is limited, so it is difficult to adapt to some large deformation movements of the human body and extract the energy generated by various human movements. However, in our manuscript, the design of the built-in helix ensures excellent tensile properties (200%) of the triboelectric yarn, as stated before, it is a major significance of our work.

(3) In Zhao, Zhizhen, et al.'s paper (Advanced Materials 28 (2016): 10267-10274, hereinafter referred to as *AM10267*), the single-layer textile TENG (t-TENG) was fabricated by plain-weaving 2-ply Cu-PET yarns as warp and PI-Cu-PET yarns as weft.

Similar to the triboelectric thread in the *AFM1604462* and the core-shell yarn in the *ACS12764*, the output performance of 2-ply Cu-PET yarns and the PI-Cu-PET yarns in the *AM10267* are also greatly affected by humidity. Besides, the 2-ply Cu-PET yarns and the PI-Cu-PET yarns are obtained by depositing Cu and coating PI on the PET substrate, so their tensile properties are limited (20%).

Most important of all, in the *AMI0267*, the single-layer t-TENG is woven using an industrial sample weaving loom. It is indeed that the weaving process of the single-layer t-TENG basically meets the requirements of industrialization, however, we have noticed that the preparation process of the 2-ply Cu-PET yarns and the PI-Cu-PET yarns is complicated. According to Experimental Section in the *AMI0267*, we roughly estimate that the preparation time of the 2-ply Cu-PET yarns and the PI-Cu-PET yarns is more than 10.5 hours. However, the normal collection speed of our triboelectric yarns can reach 28 m min^{-1} , which is significantly faster than the collection speed of the 2-ply Cu-PET yarns and the PI-Cu-PET yarns. In addition, the authors in the *ACS12764* have mentioned that the manufacturing processes of metal and polymer coatings on common yarns could be expensive and complicated for manufacturability. In comparison, it is worth noting that only commercial and low-cost materials (silicone rubber and stainless steel yarn) are used in our work. Therefore, our continuous production method is more conducive to the application of triboelectric technology in the industrial field.

(4) In Kwak, Sung Soo, et al.'s paper (ACS Nano 11.11 (2017): 10733-10741, hereinafter referred to as *ACS10733*), the stretchable TENG (S-TENG) consists of knitted PTFE fabrics for the top and bottom triboelectric layers, and knitted Ag fabrics are used for the electrode in the middle and on the back of the top and bottom triboelectric layers.

Similar to the previous three literatures, PTFE fabrics and Ag fabrics in the *ACS10733* are also exposed to open air, and the output performance of S-TENG will also be weakened by humidity. In comparison, the output performance of our triboelectric yarn is completely unaffected by humidity.

In addition, the working strain of our e-textile can reach 200%, which is far greater than the tensile strain of S-TENG (30%). Therefore, our e-textiles are more convenient for collecting biomechanical energy generated by large deformation movements of the human body.

By studying these state-of-the-art literatures, we can conclude that our work has

certain advantages in industrial preparation, underwater operation, tensile and wearable properties. We would like to emphasize those points in the revised manuscript: “First, scalable-manufacture of TENG is still very immature. In addition, the existing energy harvesting textiles are mostly fabricated by weaving (knitting, or braiding) at least two kinds of fibers (or yarns) that hold opposite electrical properties^[32-34,46]. The individual raw fibers do not show TENG behaviors. Besides, they are susceptible to humidity, and require more stringent weaving techniques in order to achieve better wearability and stretchability.”

Our revision to the manuscript:

We added the above discussions to the third paragraph of the revised manuscript. The *AMI0267* had been cited in the first manuscript (Ref. 46), and we cited three other literatures to the revised manuscript to support our work (Ref 32-34).

Corresponding changes have been marked in red in the revised manuscript.

2. The mechanism of the SETEY operation is not well presented, and the explanation is not well supported by the data. Lots of parameters that might affect the performance of TENG were mentioned in the manuscript, but they were not clearly described. For example, the author claimed that the overall output performance of SETEY was determined by yarn diameter, structure and their fabrication process, but there was no detailed analysis regarding their relations. In addition, the author expressed much on the parameters of weaving process (e.g., couple moment, torsional moment), but whether they greatly affect the performance of TENG remained unclear.

Response: Thanks a lot for your professional comments. (1) The mechanism of the SETEY operation was presented in Figure 2, Notes S7 and S8. It can be known that the factor that really affects the performance of SETEY is the contact area and separation distance between the silicone rubber tube and the stainless steel yarn at the same working frequency.

On contact area factor: the contact area is determined by the diameter of the stainless steel yarn, the inner diameter of the silicone rubber tube, and the length of the SETEY.

On separation distance factor: the separation distance is determined by the diameter of the stainless steel yarn, the inner diameter of the silicone rubber tube, and the tensile strain.

Therefore, the output performance of the SETEY is determined by the diameter of the stainless steel yarn, the inner diameter of the silicone rubber tube, the length of the SETEY, and the tensile strain.

When the silicone rubber tube is stretched outward, its length increases and the inner diameter becomes smaller, as schematically shown in Figure R1. Since the internal volume of the silicone rubber tube is always constant before and after stretching, the inner diameter of the silicone rubber tube has the following expression:

$$\pi\left(\frac{D_0}{2}\right)^2 L_0 = \pi\left(\frac{D_s}{2}\right)^2 L_s \quad (1)$$

$$D_s = D_0 \sqrt{\frac{L_0}{L_s}} \quad (2)$$

where D_0 is the inner diameter of the silicone rubber tube in the original state, L_0 is the length of the silicone rubber tube in original state, D_s is the inner diameter of the silicone rubber tube in the pre-stretching state, and L_s is the length of the silicone rubber tube in pre-stretching state. Furthermore, we find that the inner diameter of the silicone rubber tube and its pre-strain have the following relationship:

$$\varepsilon = \frac{L_s - L_0}{L_0} \quad (3)$$

$$D_s = D_0 \sqrt{\frac{1}{1+\varepsilon}} \quad (4)$$

where ε is pre-strain of the silicone rubber tube. Note that regardless of how the inner diameter of the silicone rubber tube changes, it is always larger than the diameter of the stainless steel yarn during the stretching of the silicone rubber tube. When the pre-strain (ε) is 100%, the separation distance between the silicone rubber tube and the stainless steel yarn at a tensile strain (ε_t) of 100% can be expressed as:

$$d_{(\varepsilon=100\%)} = \frac{D_s - D_y}{2} \quad (5)$$

$$d_{(\varepsilon=100\%)} = \frac{1}{2} \left(\frac{D_0}{\sqrt{2}} - D_y \right) \quad (6)$$

where d is separation distance between the silicone rubber tube and the stainless steel yarn, and D_y is diameter of the stainless steel yarn. According to Equation 6, we can get the separation distance between the silicone rubber tube and the stainless steel yarn under different parameters, as shown in Figure R2 and R4.

Since the contact area between the silicone rubber tube and the stainless steel yarn is difficult to measure, we roughly estimate the contact area by the number of helices. The number of helices under different parameters are shown in Figure R2 and R4. The relationship between the output voltage of SETEY and the diameter of the stainless steel yarn is shown in Figure R3.

In Figure S10b and S11, it can be found that the output voltage of SETEY first increases with increasing inner diameter, then decreases when the inner diameter exceeds 1.5 mm. In Figure R3, the overall output voltage of SETEY decreases as the diameter of the stainless steel yarn increases. However, when the diameter of the stainless steel yarn is less than 0.45 mm, the output performance is only slightly changed. When the diameter of the stainless steel yarn is larger than 0.45 mm, the output performance drops dramatically.

Above phenomena are attributed to the variation of contact area and separation distance between core and sheath materials. It is known that the performance of single-electrode mode TENG is greatly affected by the separation distance when the separation distance is small. However, this effect will gradually decrease as the separation distance increases [Reference: Adv. Funct. Mater. 2014, 24, 3332–3340]. Here, according to Equation 6, Figures S10b, R2, R3 and R4, we know that the performance is mainly affected by the separation distance when the separation distance is less than 0.305 mm ($D_0 = 1.5$ mm and $D_y = 0.45$ mm). When the separation distance is greater than 0.305 mm, the influence of the separation distance is weakened, and the performance is mainly affected by the contact area.

Similarly, when the length of SETEY increases, the contact area increases and the separation distance does not change (Figure S10i and S12). The performance of SETEY increases as the contact area increases. In addition, when the tensile strain changes

during the operation, the separation distance changes and the contact area does not change. When the tensile strain is less than 100%, the SETEY is in an incompletely separated state. Therefore, the separation distance is particularly small at this time, and the performance is greatly affected by the separation distance.

In summary, the overall output performance of the SETEY at the same working frequency is determined by the diameter of the stainless steel yarn, the inner diameter of the silicone rubber tube, the length of the SETEY, and the tensile strain. According to the fitting curves of Figures R5, R6, S10i and S10j, the following relationship can be obtained between them:

$$V = \begin{cases} a_3 D_0^3 + a_2 D_0^2 + a_1 D_0 + a_0, & (D_y = 0.45mm, L = 30cm, \varepsilon_t = 100\%) \\ b_1 e^{\frac{-D_y}{t}} + b_2, & (D_0 = 0.8mm, L = 30cm, \varepsilon_t = 100\%) \\ c_1 L + c_2, & (D_0 = 0.8mm, D_y = 0.45mm, \varepsilon_t = 100\%) \\ d_1 e^{\frac{\varepsilon_t}{m}} + d_2, & (D_0 = 0.8mm, D_y = 0.45mm, L = 30cm) \end{cases} \quad (7)$$

where V is the output voltage of the SETEY, L is the length of the SETEY, and ε_t is the tensile strain of the SETEY. a_3, a_2, a_1 and a_0 are the coefficients of the polynomial fit, where they are 0.37, -2.63, 4.96 and 1.93, respectively. b_1, b_2 and t are the coefficients of the single exponential decay fit, where they are $-6.56 \cdot 10^{-13}$, 4.53 and -0.0175, respectively. c_1 and c_2 are the coefficients of the linear fit, where they are 0.151 and -0.342, respectively. d_1, d_2 and m are the coefficients of the exponential fit, where they are 0.17, -0.12 and 30, respectively.

(2) In the weaving process, we mainly discussed four parameters: couple moment, torsional moment, flattening coefficient, and buckling wave height. The existence of couple moment is a normal phenomenon of a primary plied yarn, which explains why the yarn has good flexibility. Because stainless steel yarn has excellent flexibility that ordinary wire does not have, we chose it as the core yarn of SETEY.

The primary plied yarn formed by the twisting of the stainless steel yarn and the water-resistant modified polyacrylonitrile (MPAN) yarn has a large torsional moment, which makes the primary plied yarn easy to untwist. This makes it difficult for the stainless steel yarn to be tightly integrated with the MPAN, resulting in inability to efficiently collect and transfer the charge of the energy textile. However, our double-

plied yarn can avoid the adverse phenomenon of untwisting.

The flattening coefficient (η) and the buckling wave height (h_w) will directly affect the output performance of SETEY. Assuming $\eta = 0.563$ and $h_{we} = h_{wa}$, SETEY will be severely compressed in the vertical direction, thereby SETEY is difficult to be stretched during operation, and it is even difficult to effectively form helix structures inside thereof, which will seriously impair the output performance of SETEY.

Figure R1. Schematic depiction of the silicone rubber tube being stretched outward.

Figure R2. The separation distance and the number of helix vary with the inner diameter of the silicone rubber tube when the diameter of the stainless steel yarn is 0.45 mm.

Figure R3. Voltage versus the diameter of the stainless steel yarn curves measured under 100% tensile strain.

Figure R4. The separation distance and the number of helix vary with the diameter of the stainless steel yarn when the inner diameter of the silicone rubber tube is 0.8 mm.

Figure R5. The fitting relationship between the inner diameter of the silicone rubber tube and the output voltage of the SETEY when the diameter of the stainless steel yarn is 0.45 mm, the length of the SETEY is 30 cm and the tensile strain is 100%.

Figure R6. The fitting relationship between the diameter of the stainless steel yarn and the output voltage of the SETEY when the inner diameter of the silicone rubber tube is 0.8 mm, the length of the SETEY is 30 cm and the tensile strain is 100%.

Our revision to the manuscript:

We added the above discussions to Note S9. We also added “Because stainless steel yarn has excellent flexibility that ordinary wire does not have, we chose it as the core yarn of SETEY.” to Note S13. Figures R1, R2, R3, R4, R5 and R6 were added as

Figures S10a, S10d, S10e, S10g, S10c and S10f, respectively.

Corresponding changes have been marked in red in the revised manuscript.

3. This paper proposed a newly developed melt spinning approach to fabricate the core-shell triboelectric yarn, but there is no detailed characterization of the as-spun yarn. Apart from the schematic illustration, they should provide high-resolution SEM or optical images showing the longitudinal and cross-sectional morphologies of the SETEY yarn at the different stretching and releasing stages.

Response: Thanks a lot for your professional suggestion. The characterization issues you mentioned are very critical and give us a great help to revise our manuscript. In Figure R7, we provide high-resolution optical images showing the longitudinal morphology of the SETEY at the different stretching and releasing stages. The built-in helix structure always exists as the tensile strain increases before the SETEY is fully stretched (i.e., the working strain is 100%). During the stretching and releasing process, the SETEY under the same strain has almost the same longitudinal morphology, which indicates that the morphological change of SETEY has good repeatability.

However, the cross-sectional morphology of the SETEY is difficult to characterize due to the special built-in helix structure. First, as shown in Figure R8a, SETEY is a well-sealed device. Viewed from the side, we can only see its hermetically sealed part and core yarn connecting external electrode (Figure R8b), but they do not represent the built-in helix structure of the SETEY. Second, the helices at both ends of SETEY is easy to disappear if SETEY is not sealed (Figure R8c), so we can't see its true cross-sectional morphology at this time. Further, the cross-sectional morphology of the SETEY during stretching and release is more difficult to characterize. Therefore, we are so sorry that we cannot provide high-resolution optical images showing the cross-sectional morphology of the SETEY during stretching and releasing.

Figure R7. The longitudinal morphologies of SETEY under different strains during (a) stretching and (b) releasing.

Figure R8. (a) and (b) are respectively a front view and a side view of SETEY with good sealing properties. (c) Optical photo of the unsealed SETEY.

Our revision to the manuscript:

Figure R7 was added as Figure S14. Corresponding changes have been marked in red in the revised manuscript.

4. In the last session of the paper, the author incorporated SETEY yarn with polyacrylonitrile/stainless steel yarn to form the woven TENG fabric. Since the author has demonstrated that a single SETEY yarn was functionable as an amphibious wearable power supply, such SETEY yarns in principle should be able to be directly woven or knitted into a piece of highly stretchable TENG fabric. The purpose of incorporating additional yarns is not clearly illustrated.

Response: Thanks for your valuable comment. Indeed, SETEYs can be woven directly into a piece of highly stretchable TENG fabric (Figure R9a and b). But we found that incorporating them with additional yarns would be a more practical and efficient approach. The reasons are as follows. First, from a practical point of view, the deformation of the fabric is usually uniaxial stretching/contraction, and the deformation

in the other direction can be ignored (Figures 4c and R9). So weaving SETEYs in the weft direction meets the needs of practical applications. Second, the output performance can be further improved by the design of the materials when the SETEYs are woven together with other yarns. Third, fashionable textiles can be woven from the SETEYs and colorful yarns, which is especially important in the commercial applications. Fourth, in single-electrode mode, when the unit spacing of two SETEY is small, the electric field from each SETEY will interfere with each other, resulting in a decrease of the output performance for each SETEY [Reference: Adv. Funct. Mater. 2014, 24, 3332–3340]. Only when their spacing is large enough, the mutual influence will be minimized. In our TENG fabric, the warp and weft yarns are in close contact (Figure R9c), so the performance of the single SETEY is greatly weakened. As shown in Figure R10, the output voltage of B is smaller than that of A at 100% tensile strain due to mutual interference of electric fields. Besides, the weft yarn remains substantially flat when the two SETEYs are woven into a TENG fabric, as shown in Figure R9c. The warp yarn cannot be completely stretched due to the barrier of the weft yarn, thereby the output of the warp yarn (C) is smaller than the weft yarn (B), as shown in Figure R10.

Figure R9. Optical photographs of the TENG fabric woven from two SETEYs in its (a) original state, (b) tensile state and (c) enlarged view, respectively.

Figure R10. A is the output voltage of the weft yarn of the TENG fabric woven from a SETEY and a MPAN yarn. B is the output voltage of the weft yarn of the TENG fabric woven from two SETEYs. C is the output voltage of the warp yarn of the TENG

fabric woven from two SETEYs.

Our revision to the manuscript:

We added the above discussions as Note S15:

“We found that incorporating them with additional yarns would be a more practical and efficient approach. The reasons are as follows. First, from a practical point of view, the deformation of the fabric is usually uniaxial stretching/contraction, and the deformation in the other direction can be ignored (Figures 4c, S28a and S28b). So weaving SETEYs in the weft direction meets the needs of practical applications. Second, the output performance can be further improved by the design of the materials when the SETEYs are woven together with other yarns. Third, fashionable textiles can be woven from the SETEYs and colorful yarns, which is especially important in the commercial applications. Fourth, in single-electrode mode, when the unit spacing of two SETEY is small, the electric field from each SETEY will interfere with each other, resulting in a decrease of the output performance for each SETEY ^[5]. Only when their spacing is large enough, the mutual influence will be minimized. In our TENG fabric, the warp and weft yarns are in close contact (Figure S28c), so the performance of the single SETEY is greatly weakened. As shown in Figure S28d, the output voltage of B is smaller than that of A at 100% tensile strain due to mutual interference of electric fields. Besides, the weft yarn remains substantially flat when the two SETEYs are woven into a TENG fabric, as shown in Figure S28c. The warp yarn cannot be completely stretched due to the barrier of the weft yarn, thereby the output of the warp yarn (C) is smaller than the weft yarn (B), as shown in Figure S28d.”

We also added “The SETEYs can be woven with itself or with other yarns. Since the latter performs better owing to higher output and better practicality (Note S15 and Figure S28), we will discuss the latter results next.” in the revised manuscript (*part Electrical output performance of the e-textiles*). Figures R9 and R10 were added as Figure S28.

Corresponding changes have been marked in red in the revised manuscript.

5. Eventhough the SETEY itself can run under water, the weaved fabrics do not sound a stable TENG underwater. Again, the discussion and the data are quite ambiguous.

Response: Thanks for your careful comments. In this work, the weaved fabrics are a stable TENG underwater. The reasons are as follows. As shown in Figure S38a, when the energy textile (e-textile) is short-circuited in water, the electrode pattern will shift from warp-weft-connection single-electrode pattern to weft-connection single-electrode pattern. This auto-shift operation mode does not affect the contact-separation state inside the SETEY, and the e-textile of the weft-connection single-electrode pattern can still operate normally. As shown in Figure R11, the output performance of the e-textile of the weft-connection single-electrode pattern underwater is superior to that in the air (Figure 4d) due to the potential-polarization coupling effect, and output performance of the e-textile is also stable during the tensile test.

Figure R11. The output voltage of the e-textile of the weft-connection single-electrode pattern underwater when the tensile strain is 100%.

Our revision to the manuscript:

We added “As shown in Figure S38b, the output performance of the e-textile of the

weft-connection single-electrode pattern underwater is superior to that in the air (Figure 4d) owing to the potential-polarization coupling effect, and output performance of the e-textile is also stable during the tensile test.” in the revised manuscript (*part Electrical output performance of the e-textiles*). Figures R11 was added as Figure S38b.

Corresponding changes have been marked in red in the revised manuscript.

Thank you again for your great help & Merry Christmas.

REVIEWERS' COMMENTS:

Reviewer #1 (Remarks to the Author):

I appreciate all the efforts the authors have made in order to improve the quality of the manuscript. All the questions/comments from me have been well responded. I have no more questions/comments.

Reviewer #2 (Remarks to the Author):

The authors have addressed the comments made by the reviewers adequately.

Reviewer #3 (Remarks to the Author):

I really appreciate the author's effort in providing a very detail response to the comments. But as all the reviewers' pointed out, although the work is very good and indeed interesting, it does not reach the standard novelty level of Nat Commun papers. That is an issue that difficult to address from the response.